

# Evaluation of the Moisture Sources in two Extreme Landfalling Atmospheric River Events using an Eulerian WRF-Tracers tool.

Jorge Eiras-Barca[1], Francina Dominguez[2], Huancui Hu[2], A.Daniel Garaboa-Paz[1], and
Gonzalo Miguez-Macho[1]

[1]Non-Linear Physics Group, Universidad de Santiago de Compostela, Galicia, Spain.
[2]Departament of Atmopsheric Sciences, University of Illinois at Urbana-Champaign, IL.

*Correspondence to:* Jorge Eiras Barca (jorge.eiras.b@gmail.com)

**Abstract.**

A new 3D Tracer tool is coupled to the WRF model to analyze the origin of the moisture in two extreme Atmospheric River (AR) events: the so-called "Great Coast Gale of 2007" in the Pacific Basin, and the "Great Storm of 1987" in the North Atlantic. Results show that between 80% and 90% of the moisture advected by the ARs, as well as between 70% and 80% of

the associated precipitation have a tropical or subtropical origin. Local convergence transport is responsible for the remaining moisture and precipitation. The ratio of tropical moisture to total moisture is maximized as the cold front arrives to land. Vertical cross sections of the moisture suggest that the maximum in humidity does not necessarily coincide with the Low-Level Jet (LLJ) of the extratropical cyclone. Instead, the amount of tropical humidity is maximized in the lowest atmospheric level in southern latitudes, and can be located above, below or ahead the LLJ in northern latitudes in both analyzed cases.

# 1   Introduction

Atmospheric Rivers (hereafter, ARs) are long and narrow structures in the lower troposphere that carry large amounts of water vapor (Zhu and Newell, 1998). Guan and Waliser (2015) estimate that ARs have a median length of about 3600 km, a median length/width ratio of about 7 and mean water vapor transport (IVT) of $370 \, \mathrm{kg \cdot m^{-1} \cdot s^{-1}}$. ARs transport moisture towards the Warm Conveyor Belts (WCBs) of extratropical cyclones by thermal wind advection in the low-level jet -a region of strong

winds located in the lower part of the atmosphere, generally linked to cold fronts- (Dettinger et al., 2015). Nevertheless, in practice, the relationship between ARs and cyclones has been documented to be complex (e.g. Sodemann and Stohl, 2013). Between 3 and 5 ARs can be found per hemisphere at any given time, accounting for approximately 84% of the meridional IVT for the Northern Hemisphere and about 88% in the Southern Hemisphere (Guan and Waliser, 2015).

Since these structures can transport an amount of precipitable water equivalent to several times of the discharge of the Missis-

sippi River (Ralph and Dettinger, 2011), ARs have been identified as a primary feature of the global water cycle. Furthermore, ARs are considered as one of the potential precursors of extreme precipitation, particularly when landfall occurs (e.g. Gimeno et al., 2016). The relationship between ARs and flood events has been extensively analyzed for the U.S. West Coast Region (e.g. Higgins et al., 2000; Ralph et al., 2005; Bao et al., 2006; Ralph et al., 2006; Neiman et al., 2008a, b; Leung and Qian,



2009; Dettinger, 2011; Dettinger et al., 2011; Warner et al., 2012; Rutz et al., 2013; Kim et al., 2013; Ralph et al., 2004, 2013; Warner et al., 2014), Europe (Lavers et al., 2011, 2012; Lavers and Villarini, 2013; Lavers et al., 2013; Lavers and Villarini, 2014; Ramos et al., 2015; Eiras-Barca et al., 2016; Ramos et al., 2016; Brands et al., 2016), and other regions worldwide (e.g. Mahoney et al., 2016; Mundhenk et al., 2016). It is important to better understand the physical mechanisms leading to extreme

flooding associated to ARs, considering their impacts on human and natural systems and the mounting evidence that ARs are projected to become more frequent and intense in the future (Dettinger, 2011; Lavers et al., 2013; Payne and Magnusdottir, 2015, e.g.).

There are several proposed methods of AR detection, most of which are based on thresholds of integrated water vapor (IWV) and/or integrated water vapor flux (IVT), shape criteria, and from satellite or reanalysis data (e.g. Ralph et al., 2004;

Bao et al., 2006; Lavers et al., 2011; Dettinger, 2011; Lavers et al., 2012; Nayak et al., 2014; Ramos et al., 2015; Eiras-Barca et al., 2016; Brands et al., 2016). Guan and Waliser (2015) has developed a global detection method using filters of intensity, direction, geometry and coherence of the structures. More recently, Eiras-Barca et al. (2016) proposed a combined IVT and IWV variable-threshold detection algorithm, which operates both in summer and winter months. These objective detection criteria have shown that AR structures of IWV and IVT can extend from the tropics and subtropics into the mid latitudes;

however, they do not provide information about the source and sink regions of the AR water vapor.

Tropical moisture exports (TME) have been identified as a primary source of moisture for ARs in Europe and the U.S. West Coast. AR structures link remote sources of moisture from the (sub)tropics to mid-latitudes through long corridors of advection (e.g. Knippertz and Wernli, 2010; Sodemann and Stohl, 2013; Knippertz et al., 2013; Ryoo et al., 2015; Ramos et al., 2015). These studies have primarily used backward Lagrangian tools to evaluate the source-sink regions. However, moisture

from mid-latitudes (local sources) has also been identified as an important source of water vapor convergence in AR events (Dettinger et al., 2015). Ramos et al. (2016) used the FLEXible PARTicle dispersion model (FLEXPART) to show that both tropical and local sources of moisture are present in AR landfall events for different European latitudes. While the idea that both local and remote sources contribute to ARs, some authors argue that local sources are primarily responsible for the high water vapor content within the AR-core (Bao et al., 2006; Cordeira et al., 2013; Dacre et al., 2014). By calculating the water

vapor budget of 200 extratropical cyclones, Dacre et al. (2014) conclude that tropical moisture reaching the extratropics is only contributing to mid-level moisture, above the boundary layer. Under this perspective, ARs can be thought as the footprints left behind the cyclone pathway, and not as a conduit for meridional transport of water vapor and latent heat. One possible explanation for the lack of agreement may be the sensitivity to the physics and parametrization schemes used in the latter analysis (Brands et al., 2016).

Considering that there is still an important discussion related to the origin of water vapor in ARs, in this paper, we use a new forward moisture tracer tool coupled to the Weather Research and Forecast (WRF) model (Miguez-Macho et al., 2013; Dominguez et al., 2016) to evaluate the moisture sources of two particularly extreme AR-case studies, as well as the transport mechanism of this humidity. The first AR developed over the Pacific Ocean and affected the west coast of North America, whereas the second AR developed over the Atlantic Ocean and impacted the west coast of the Iberian Peninsula. The tracer

tool allows us to track the tropical moisture associated with these two events, and evaluate the relative contribution of this



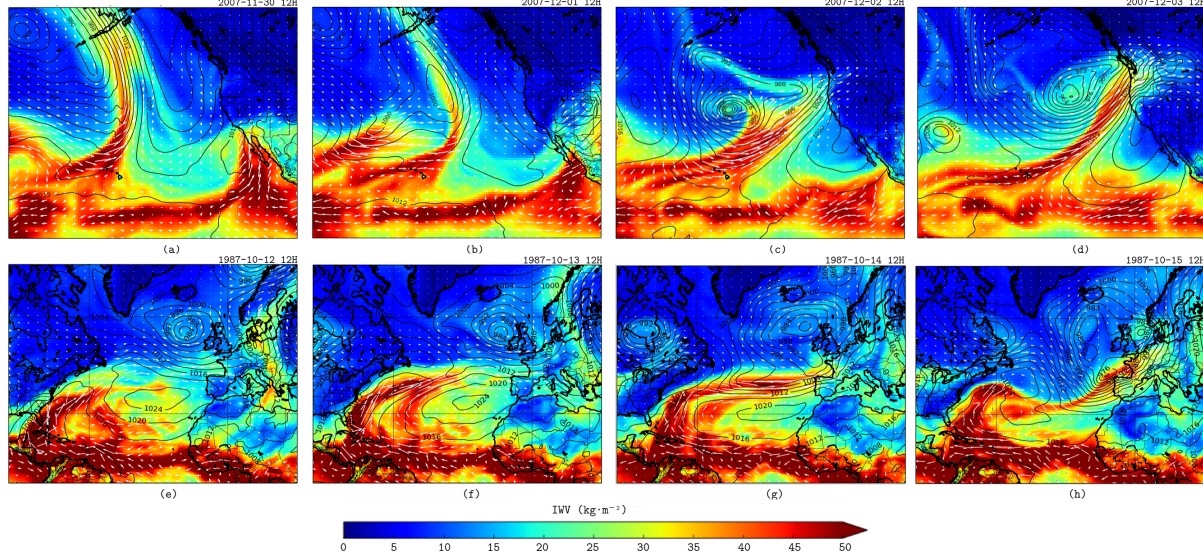

**Figure 1.** Integrated vapor transport (IVT, vectors, $\mathrm{kg \cdot m^{-1} \cdot s^{-1}}$), sea level pressure (SLP, isobars, hPa) and integrated water vapor (IWV, background, $\mathrm{kg \cdot m^{-2}}$) fields for both "the Great Coast Gale" (a-d) and "the Great Storm" (e-h) events troughout a four-days window time frame. Source: Era-In.

tropical moisture to total water vapor and precipitation. In addition, the WRF tracer tool also provides information about the vertical distribution of tropical moisture, as well as the position of the maximum of moisture with regard to the low-level jet. This manuscript is organized as follows. Section 2 describes the applied data and methods, the results and discussion are presented in section 3 and we summarize our conclusions in section 4.

## 2   Methods

We select two AR events that resulted in extreme precipitation and flooding, causing significant socioeconomic damages. The first AR occurred in December of 2007, affecting mostly the Pacific North West region of the United States [Figure 1, a]. Locally known as the "Great Coastal Gale", this event primarily impacted the western state of Washington and the associated flooding resulted in approximately $680 million direct losses from three severely impacted counties in the state (Dall'erba, 2016) and 11 fatalities (NOAA, 2008). Formed from the remnants of the two typhoons Hagibis and Mitag, the event lead to hurricane force warnings (Crout et al., 2008). The rapid -explosive- development of the cyclone is shown in Figure 1.a. The selected event was the third and most intense of a series of three storms, and lead to extreme precipitation due to the strong linkage to the only Atmospheric River of the Gale (NOAA, 2008). The landfalling event and the resulting precipitation associated with the cyclone and the atmospheric river is shown in the Figure 2.a.

The water vapor signature of the second event, which developed in October of 1987, extended from the western tropical Atlantic Ocean to the Iberian Peninsula. This event is the well-known "Great Storm of 1987", with reported losses of millions





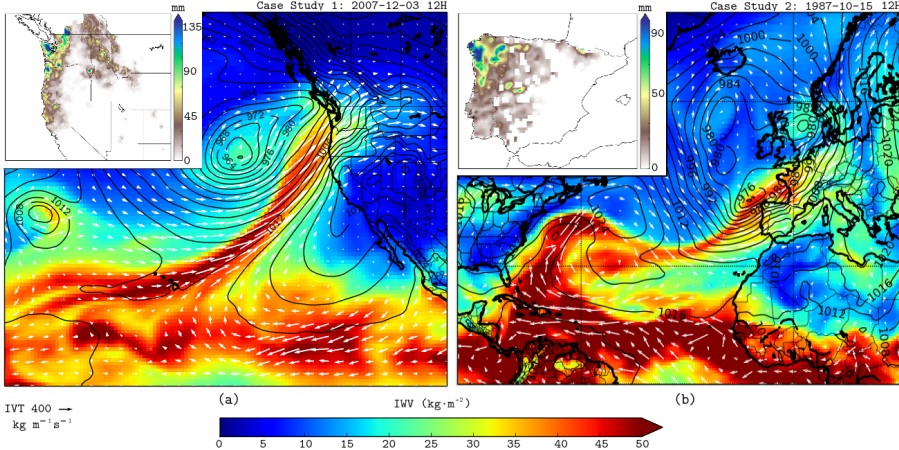

**Figure 2.** Same as Figure 1 but only the precise moment when the landfall of the atmospheric river occurs is shown. Upper-left boxes show the 24-hour accumulated precipitation in mm.

of pounds and 18 fatalities over the British Isles (e.g. Burt and Mansfield, 1988). Regarding the alleged role of the atmospheric river in the fast deepening of the cyclone -35 mb in 24 hours- (Figure 1, b), Shutts (1990) showed the key role played by the latent heat release in the storm formation. This analysis claims that two thirds of the central pressure falling could be ascribed to latent heat release. Figure A1 in supplementary material show how the cooperative linkage between a trough in the tropopause and low-level baroclinicity have contributed to the rapid growth of the system as well (Hoskins and Berrisford, 1988). The resulting precipitation was reported above 100 mm throughout the Spanish region of Galicia, shown in Figure 2.b.

We use the Weather Research and Forecasting Model (WRF 3.4.1) to simulate these two events. For the Pacific case, the WRF horizontal resolution is 15 km and the vertical column is divided into 40 levels. For the Atlantic simulation, grid spacing is 20 km in the horizontal and there are 50 vertical levels. Both simulations cover a period of 10 days, from November 26th, 2007 in the Pacific case and from October 8th, 1987 in the Atlantic simulation. The Water Vapor Tracer (WVT) tool has been implemented in the YSU planetary boundary layer parameterization (Hong et al., 2006; Shin and Hong, 2011; Hu et al., 2010, 2013), the Kain-Fritsch convection scheme (Kain, 2004) and the WSMC6 microphysics scheme (Hong and Lim, 2006), which are the paramerizations employed in the simulations; in addition, the RRTM (Mlawer et al., 1997) and Dudhia (Dudhia, 1989) schemes were used for long wave and short wave radiation respectively. Spectral nudging has been applied to avoid distortion of the large scale circulation within the regional model domain due to the interaction between the model's solution and the lateral boundary conditions (Miguez-Macho et al., 2004, 2005). Further descriptions about WRF can be found in Skamarock et al. (2005) or Michalakes et al. (2005). Finally, and considering that the ECMWF reanalysis (ERA-Interim) has been shown to be a reliable tool in the analysis of ARs (Rutz et al., 2014), this dataset provides lateral boundary and initial conditions for the runs.

Since spectral nudging has been used in the simulations, the large scale circulation in the model closely follows ERA-Interim and no further validation is required (Gómez and Miguez-Macho, 2017). Water vapor is not nudged, and given that





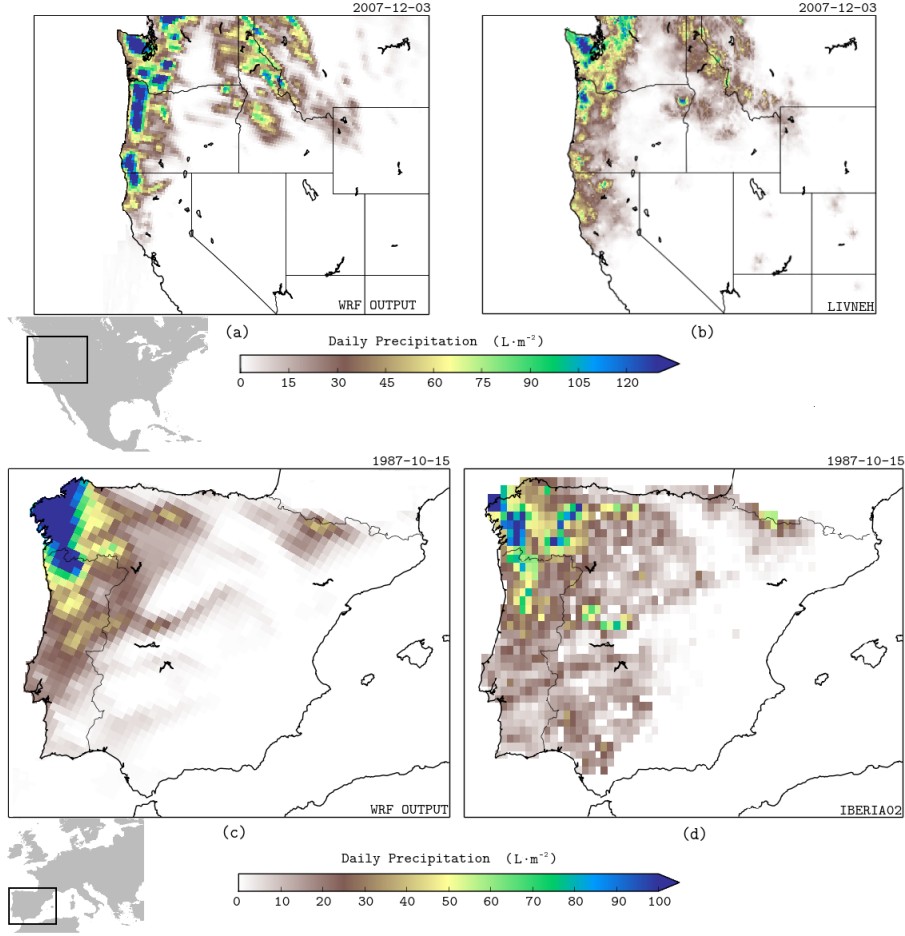

**Figure 3.** WRF Output total precipitation for the "Great Coast Gale" (a) and the "Great Storm" (c) against observations from LIVNEH (b) and IBERIA02 (d), for the same 24-h period.

the subject of this study is moisture transport and precipitation, we focus validations on these two variables. Figure 3 shows the comparison between WRF-simulated and observed precipitation. Observations are from the Livneh et al. (2015) dataset for the Pacific simulation and from the Iberia02 precipitation dataset in the case of the Atlantic simulation. The latter dataset is a combination of Spain02 (Herrera et al., 2012) and Portugal02 (Belo-Pereira et al., 2011), both of which include a high density of stations of good reliability (Herrera et al., 2012). Further comparison of the simulations with observations is provided in [FigVALQ], against Integrated Water Vapor (IVT, eq. 1) from NASA's Modern-Era Restrospective Analysis for Research and Aplications (MERRA) (Rienecker et al., 2011).

Whereas the simulated IVT field is realistic when compared to observations, WRF tends to overestimate precipitation. The overestimation is particularly high in the mountains of Oregon and Washington for the 2007 event. However, despite the fact





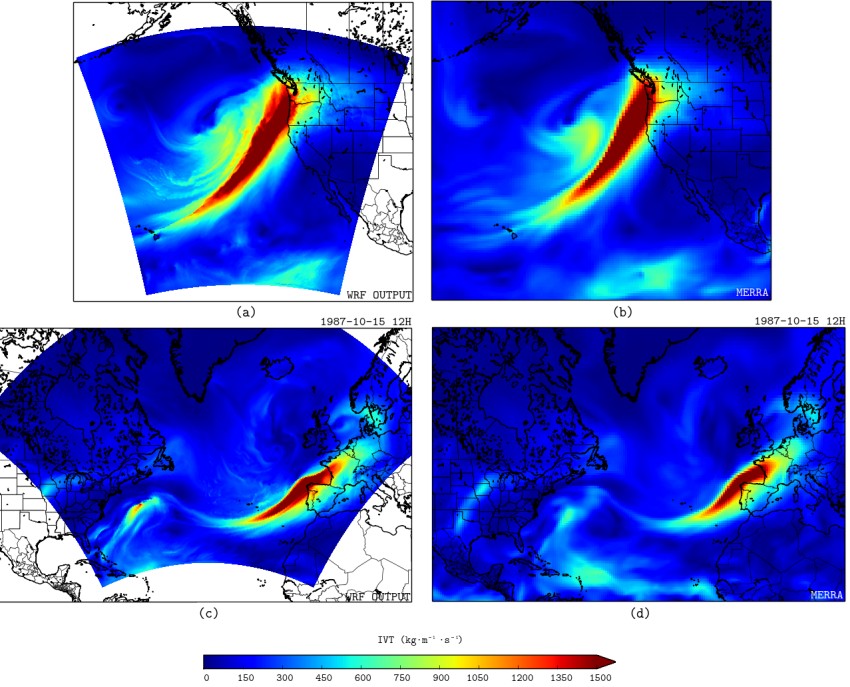

**Figure 4.** Integrated vapor transport (IVT) in $kg \cdot m^{-1} \cdot s^{-1}$ for the Pacific event from WRF (a) and MERRA (b) as well as for the Atlantic event from WRF (c) and MERRA (d).

that precipitation is known to be the most difficult parameter to simulate in a numerical model (e.g. Maraun et al., 2010; Buckley and Marshall, 2016), the spatial pattern of precipitation is realistically represented.

The Eulerian tracer tool operates as follows: A wide region in the domain covering the tropical latitudes is set up as a three-dimensional tracer mask. Notably, while previous water vapor tracer configurations in WRF focused on tracking water that

5    evaporated from a two-dimensional region at the surface (e.g. Dominguez et al., 2016), in this study all the water vapor in a three-dimensional volume (including the water vapor evaporated and advected into the masked region) is tracked in space and time. Figure 5 shows the masks labeled in red for the Pacific (a) and Atlantic (b) simulations. Once the simulation starts, the model tracks the humidity originating from within the mask at any time, and the quantity of this moisture is known in relation to the total moisture content in each point of the domain, throughout the entire simulation. Similarly to the rest of the moisture,

10    the "tagged" water vapor can change phase, and the fraction of the condensed phase to the total condensate is also reported in the model output.

Finally, the Integrated Column of Water Vapor (IWV), and the Integrated Column of Water Vapor Tracers (IWV$_{\mathrm{TR}}$) can both be calculated from the WRF simulations using equations 2 and 3 respectively; where $q$ is the specific humidity, $g$ is gravity,





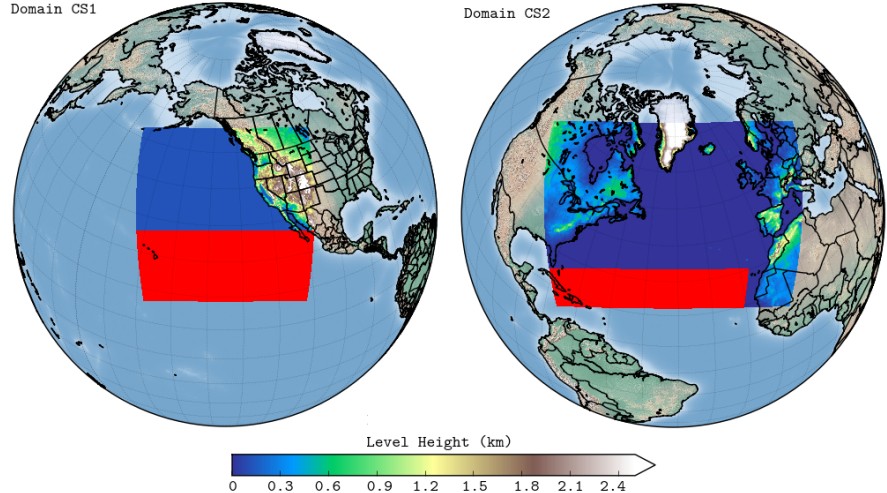

Level Height (km)

0    0.3   0.6   0.9   1.2   1.5   1.8   2.1   2.4

**Figure 5.** Domains of the WRF simulation for the "Great Coast Gale" (a) and the "Great Storm" (b). Areas highlighted in red correspond to the mask region which is where the moisture is initially labeled as tracer.

$u$ and $v$ represent the wind field an $l$ is the highest model level, well above the tropopause. The conversion between specific humidity ($q$) and mixing ratio ($w$) has been done using equation 4.

$$IVT = \left| \frac{1}{g} \int_{sfc}^{l} q\mathbf{u}dp \right| \tag{1}$$

$$IWV = \frac{1}{g} \int_{sfc}^{l} qdp \tag{2}$$

$$IWV_{TR} = \frac{1}{g} \int_{sfc}^{l} q_{TR}dp \tag{3}$$

$$q = \frac{w}{w+1}, \text{ with } w \ll 1 \Rightarrow q \approx w \tag{4}$$

## 3   Results and discussion

Figure 6 shows the three-dimensional distribution of water vapor mixing ratio (a), and tracer water vapor mixing ratio (b) for the event in the Pacific that made landfall along the U.S. West Coast on December 3, 2007. While the former accounts for the





total amount of moisture, the latter shows only the moisture originating from (sub)tropical latitudes, labeled with the 3D mask depicted in Figure 5. The simulation was started 8 days before the time shown in Figure 6. Pannels (c) and (d) show a snapshot of the water vapor mixing ratio and the tracer water vapor mixing ratio in the form of a series of cross-section slices that allow the visualization of the vertical distribution of moisture. The images suggest that the vast majority of the moisture contained

in the pre-frontal region has its origin in the tropical regions. This is especially true at lower latitudes. The maximum content of tropical moisture remains mostly in the lower levels. The local convergence mechanism seems to take place only behind the frontal region, as well as along the leading edge of the AR structure, where the WCB is located. Slightly different conclusions are obtained for the Atlantic case study shown in Figure 7 (For the sake of simplicity, only the eastern longitudes of the domain of simulation are shown in the Figure). Even though the tropical moisture still remains in the lowest levels of the troposphere,

its contribution to the total is less significant than for the Pacific case study. The reason for the latter is shown in Figure 1(e,f,g and h), indicating that most of the connection of the WCB with tropical regions is through a much longer path, due to the blocking position of the Azores High. The main goal of Figures 6 and 7 is the visual depiction of the total and tracer moisture.

Figure 1 shows the development of both the Pacific and Atlantic events, during the four days prior to landfall. Figure 1.c and Figure 1.h depict how both cyclones undergo explosive cyclogenesis (the central pressure falls more than $24 \cdot sin\varphi/sin(60°)$

hPa in 24 hours). The December 2007 AR in the Pacific developed from the merger of a cold front, already undergoing wave development, and a faster moving low pressure system catching up from behind. Both systems had been in origin typhoons Hagibis and Mitag and already had a high water vapor content. The interaction between both resulted in an instant occlusion-like mechanism that lead to the rapid deepening of the combined cyclone over the Pacific. Figure 1.h shows the explosive cyclogenesis for the European event. In this case, there is also a complex development process, with the interaction between

the remnants of a tropical system with high water vapor content and a wave on the long frontal boundary across the North Atlantic as precursor of the explosive cyclogenesis occurring north west of the Iberian Peninsula (see Figure A2).

Figure 8.a shows the percentage of IWV that comes from the tropics ($100 \cdot IWV_{TR}/IWV$) for the Pacific December 2007 event. In a region extending from the tropics into the coast of the state of Washington, tropical moisture accounts for more than 90% of the precipitable water in some points. These high percentages extend inland along the Northwestern U.S. coastal

regions. Likewise, Figure 8.b shows 24h-accumulated percentage of precipitation that is composed of condensed tropical water vapor ($100 \cdot Prec_{TR}/Prec$). For clarity, we only plot the region where precipitation exceeded 3mm (Buishand, 1978). Precipitation of tropical origin accounts for 70% to 90% of total precipitation in northern California and southern Oregon, and the ratio decreases at higher latitudes. Interestingly, tropical moisture is funneled by local topography, and it contributes to about 70% to 80% of precipitation west of the Cascade Mountain Range. In the October 1987 Atlantic case, we also see a

clear plume where tropical water vapor accounts for more than 80% of precipitable water; however, the percentage decreases rapidly to around 70% before arriving on the Iberian coast (Figure 9). Precipitation, consequently is only between 70% to 80% of tropical origin.

Figure 10 plots a range of transversal cross sections showing the vertical distribution of tracer water vapor mixing ratio through the central axis of the AR, as well as wind speed. In the figure, there is evidence that the maximum of tropical moisture

does not necessarily coincide with the low-level jet (LLJ), which is the maximum in wind speed at lower levels. Precisely, at





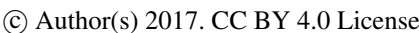

**Figure 6.** a) Total water vapor mixing ratio in g/kg at 2007-12-03 12hUTC for the Pacific domain. b) Tracers water vapor mixing ratio in g/kg at the same time and domain. c) Vertical cross sections of (a). d) Vertical cross sections of (d).

the root of the AR, in subtropical latitudes (Figure 8.d, most of the tropical moisture remains close to surface and below the LLJ, which can be identified at a height of 1 km. As the central axis of the LLJ goes upward with latitude, tropical moisture tends to ascend in the vertical column, but the maximum of moisture can be located in front or behind the LLJ; as well as remaining in near surface levels. In the leading part of the AR, the interaction of the humidity with the topography of the Pacific Coast of North America makes the situation more difficult to analyze. Very likely, the complex formation process of the cyclone, from the interaction of two preexistent frontal systems, loaded with tropical moisture, adds complication to the thermodynamic structure and moisture distribution of the resulting front. Notwithstanding, this AR event is a particularly well defined case from the perspective of vertically integrated quantities such as IVT and IWV.





**Figure 7.** Same as Figure 6 but for the European domain in the "Great Storm" (1987-10-15 12hUTC).

An analogous plot for the Atlantic case is presented in the supplementary material (Figure A2). The results are similar to the Pacific case, with no clear one-to-one connection between the LLJ and the maximum in tropical humidity. We note that no general conclusion may be obtained from particular case studies, but these results suggest that the perception that ARs are clearly associated with the LLJ of extratropical cyclones should be reviewed.

Figure 11.a shows the area-averaged total precipitation (black circles) and the ratio between tracer precipitation and total precipitation (red crosses) throughout the region highlighted in Figure 11.b for the Pacific event. As expected, the plot shows



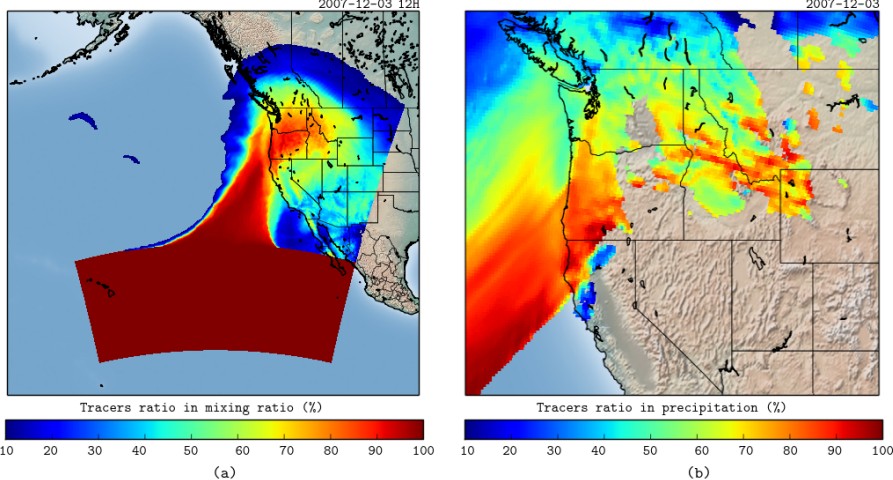

**Figure 8.** a) Tracers ratio in mixing ratio (g/kg) out of the total moisture mixing ratio (g/kg) for the Pacific event . This value quantifies the ratio of tropical precipitation with regard to the total. b) Same as (a) but for precipitation.

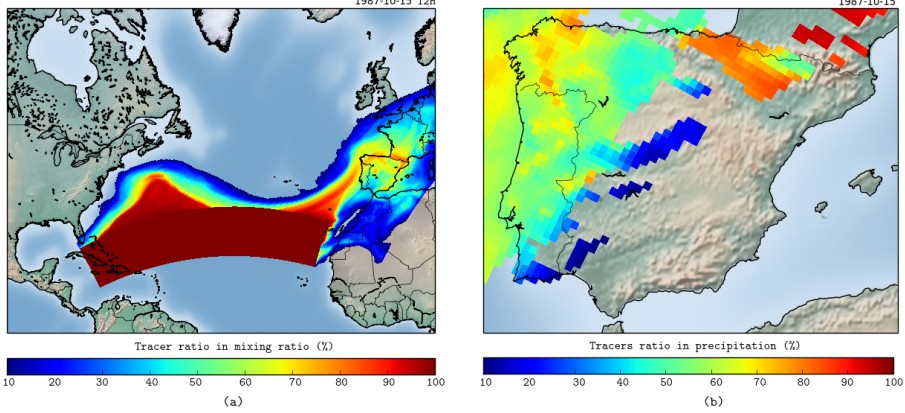

**Figure 9.** Same as Figure 8 but for the Atlantic domain.

that the maximum in tropical precipitation is observed during the landfall of the cold front and the AR, which closely coincides with the maximum in total precipitation. The secondary maximum in the total precipitation observed one day before the landfall of the AR event is due to the landfall of the warm front associated with the cyclone (see Figure A3). The convergence of local moisture would be the dynamical source for precipitation in the warm front.



**Figure 10.** Transversal cross sections along the central axis of the atmospheric river at latitudes 42.0 (TCS P1), 37.4 (TCS P2) and 30.6 (TCS P3). The plots show the tracers water vapor mixing ratio in g/kg together with the wind module in m/s. The estimated position of the low level jet is pointed in the figures as well.

## 4 Conclusions

A new 3D Eulerian forward water vapor tracer tool implemented in the Weather Research and Forecasting (WRF) model has been used to analyze two important atmospheric river events. The first event developed over the Pacific Ocean and corresponds to the "the Great Coast Gale of December, 2007" in the Pacific U.S. west coast, which caused several million dollars in direct economic damages. The Atlantic event corresponds an atmospheric river event in October 1987 that resulted in record winds of 100 km/hr and daily precipitation over 100mm in Galicia (in the northwest of Spain), and Portugal. In an effort to understand



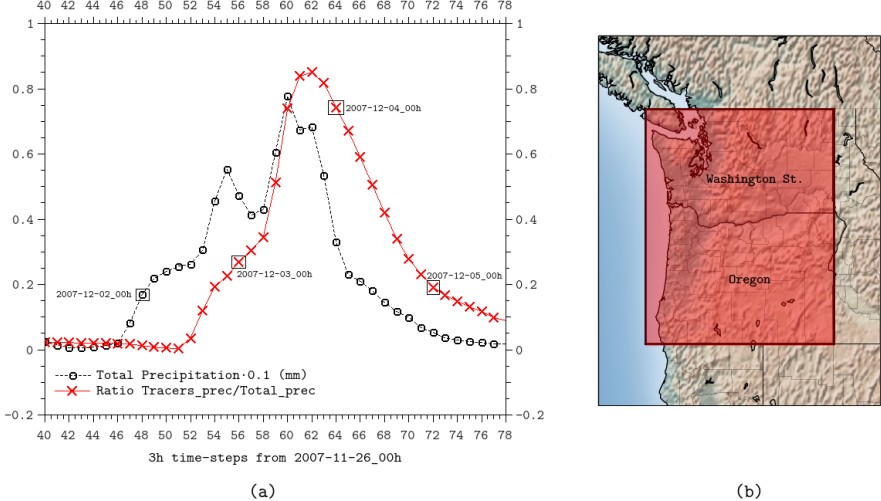

(a)                                                                  (b)

**Figure 11.** Evolution over time of the tropical precipitation (red crosses) and total precipitation (black circles) during the Pacific event (a). Data represent the spatial integration of both variables through the region highlighted in red in (b).

the origin of moisture for these two AR events, we use 3D water vapor tracers to quantify the percentage of total precipitable water and precipitation that originates from the tropics.

Results show that most of the moisture within and surrounding the two atmospheric rivers had its origin in the tropical regions that we labeled with the 3D tracer mask. Consequently, most of the precipitation that fell during these two events

5    was composed of condensed tropical water vapor. The Pacific event shows a more intense connection with tropical regions; therefore, the percentage of tropical precipitation for this event is higher and peaks at around 85%. Nevertheless, even for the October 1987 Atlantic event, more than 60% of the resulting precipitation is of tropical origin.

The two selected case studies have been chosen in terms of heavy precipitation, and both correspond to a strong atmospheric river feeding the system of a very intense extratropical storm. The conclusions drawn from these two AR events are thus

10   not necessarily representative of the bulk of Pacific or Atlantic AR events. The results highlight, however, the importance of tropical moisture for the two case studies. We also find evidence that convergence of local moisture also contributes to total precipitable water, especially in the post-frontal region, the leading edge of the AR and in the far northern latitudes where the tropical link has weakened. It is well known that in a mature system, the water vapor store tends to be constant (e.g. Bullock and Johnson, 1971), and since the fate of tropical moisture is to precipitate sooner of later, local convergence should keep the

15   balance by lateral inflow.

Based on these results, we hypothesize that the highest amounts of precipitable water can only be attained in a system when a clear tropical source of moisture that is sustained until the system makes landfall. Strong ARs with a direct link to tropical latitudes should be expected to result in more precipitation than those with local convergence as a primary feeding mechanism. It is our aim for the future to extend this work by including a larger amount of cases.



Finally, our findings suggest that the maximum of tropical moisture does not necessarily coincide with the low-level jet of the extratropical cyclone in neither case analyzed. Instead, this maximum is located in near surface levels at lower latitudes to later ascend to levels aloft in northern latitudes, but not within the LLJ. The maximum of tropical moisture may be situated below, behind or in front of the LLJ, which is located along the cold front. Both events are very clear examples of ARs from the
5   point of view of vertically integrated variables, such as IWV and IVT used in most detection algorithms; however the vertical distribution of moisture, mostly of tropical origin, reflects the complex processes leading to their explosive cyclogeneses. In both cases, remnants of tropical systems were involved as precursors. It is widely accepted in the literature that the bulk of moisture in ARs is primarily advected within the LLJ of extratropical cyclones but in light of our results we suggest that further discussion is necessary for this matter.





# Appendix A: Supplementary Figures

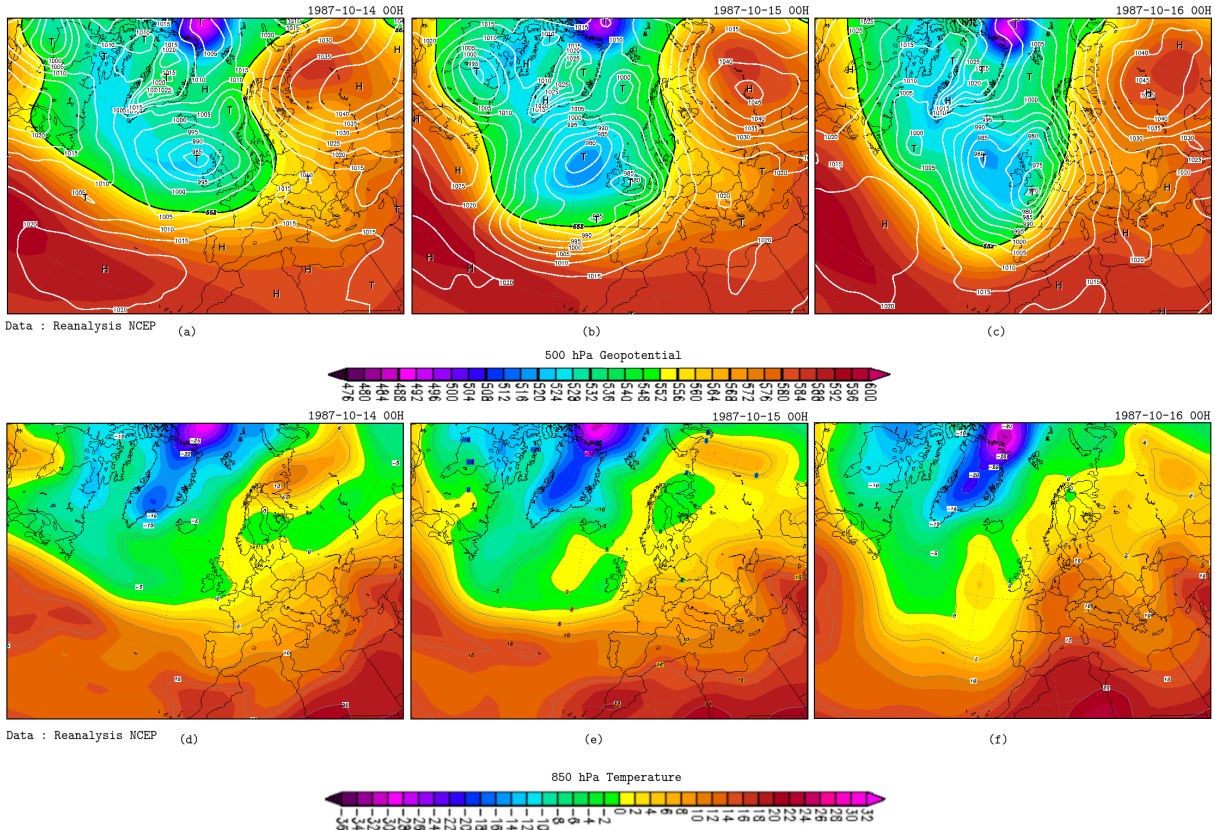

**Figure A1.** 500 hPa Geopotential field together with SLP (a-c) and Temperature (d-f) in the Great Storm of 1987 event from October 14th to October 16th. The figure highlights the cooperative linkage between between a through and low-level baroclicinity in the rapid development of the cyclonic system.





**Figure A2.** Same as Figure 10 but for the european case. The corresponding latitudes are 41.8 for TCS A1 and 38.0 for TCS A2.

*Acknowledgements.* This work has been founded by the Ministerio de Economía y Competitivad (CGL2013-45932-R) from the Spanish Government and its mobility grants for pre-doc researchers. Jorge Eiras-Barca would like to express his gratitude to the Department of Atmospheric Sciences of the University of Illinois at Urbana-Champaign for the kind support in this project. Funding for Dominguez and Hu comes from the U.S. National Aeronautics and Space Administration (NASA) Grant NNX14AD77G.



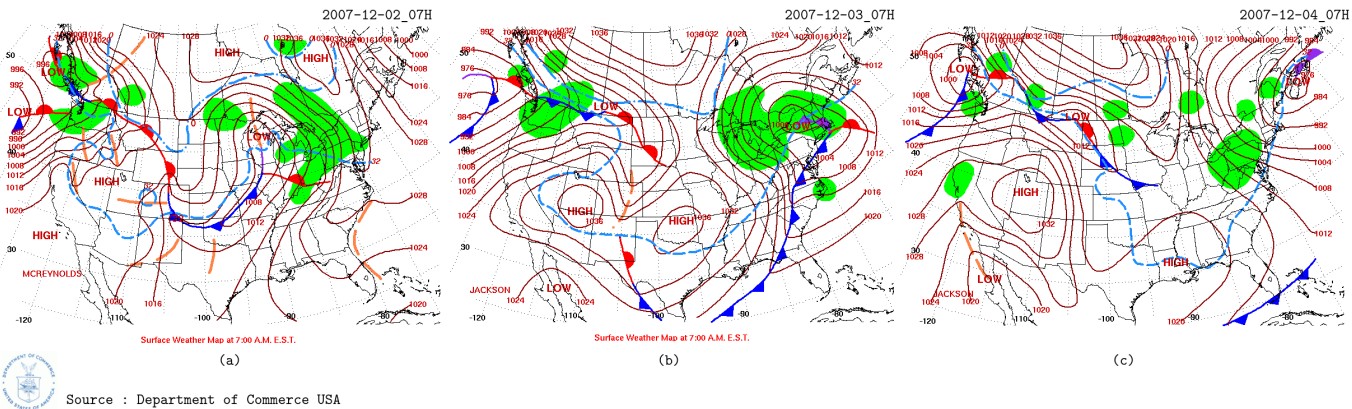

**Figure A3.** Front maps for the Pacific "Great Coast Gale" event.

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
