# Peer review of "Evaluation of the Moisture Sources in two Extreme Landfalling Atmospheric River Events using an Eulerian WRF-Tracers tool."

_Earth System Dynamics, 2017_

## Referee Comment (RC1) · R. J. van der Ent (Referee) · 12 Jul 2017

**Referee comment of Eiras-Barca et al in Earth System Dynamics**

**General comments**

This paper investigates the (atmospheric) moisture sources of the "atmospheric river" events. One event on the western coast of North America and another event on the western coast of Europe. The authors implement a relatively new and novel online moisture tracing tool in WRF, which allows them to question mark the apparent previous consensus that atmospheric rivers are associated with the low-level-jet of extratropical cyclones. This contribution is certainly valuable to this special issue of Earth System Dynamics, and I do not have any major comments. However, I do have quite a lot of minor comments mostly regarding readability due to imprecise use of terminology and symbols, overuse of acronyms, as well as several unclarities in the figures.

**Specific comments**

It would be useful if the authors could think of a name for the WRF moisture tracing tool

P1, L2-L5: "the so-called "Great Coast Gale of 2007" in the Pacific Basin, and the "Great Storm of 1987" in the North Atlantic. Results show that between 80% and 90% of the moisture advected by the ARs, as well as between 70% and 80% of the associated precipitation have a tropical or subtropical origin."
I was intrigued by this statement and wondered about the different between atmospheric moisture and precipitation, however, it seems that the percentages are quite coarsely estimated from visual inspection (numbers for Canada would be much lower) and no hard conclusions can be drawn from this. I wonder whether more deterministic percentages could be calculated for when the AR event makes landfall or for precipitation occurring within x km from the coast during x days of the AR event. Moreover, the word subtropical does not come back anywhere in the paper. Why? Is it wrong in the abstract or in the rest of the paper?

P1, L8: it makes no sense at all to abbreviate 'mean water vapor transport' with IVT, moreover, the authors are not very consequent as on page 2 it suddenly appears that IVT stands for integrated water vapor flux. Judging from the acronym a logical term would be 'integrated vapor transport' or the acronym should be changed.

P2, L17: " … West Coast"
Reference?

P2: "By calculating the water vapor budget of 200 extratropical cyclones, Dacre et al. (2014) conclude**d** that tropical moisture reaching the extratopics is only contributing to mid-level moisture, above the boundary layer."
I think the authors find something else, it would be nice if they could reflect on this statement in their conclusions.

Figures in general: Use clear headings instead of tiny names in the figure corners. The figures that show both AR events would benefit from clear titles specifying which panels are referring to the "Great Coast Gale" and the "Great Storm" respectively.

Figure 1: The source should be spelled out ERA-Interim instead of ERA-In. "SLP" appears only 2 times in the paper, thus no need to abbreviate this. A scale of the IVT vectors is missing.

P3, L1: "total water vapor"
Does the tool also track liquid water and ice and the associated phase transitions? Please elaborate on this and the consequences for the results if the assumption is that it only tracks vapor. On the other hand the authors should adjust their terminology if the tool in fact includes liquid water and ice.

P3, L3: "data and methods"
Then call section 2 data and methods instead of just methods

P3, L3: "summarize our conclusions"
After reading section 4 it appears that you give a summary and conclusions, which is different from summarizing conclusions.

P3, L16: "Iberian Peninsula"
Later you refer to damages on the British Isles, thus the AR appears to extend further than the Iberian Peninsula.

Section 2: subheadings would be helpful for readability

P4: "YSU", "WSMC6", "RRTM"
What do these acronyms mean? The way they appear now they are not helpful for readability.

P4, L20: "spectral nudging"
Can the authors be more specific, as there are many ways to apply spectral nudging.

Figure 3: What is ($L\ m^{-2}$)? This would read as Luminous intensity per square meter according to the International System of Units, which I hardly think the authors mean. The total precipitation is shown on land only, be specific about this in the caption. LIVNEH and IBERIA02 do not have to be capitalized.

P5, L6: "[FigVALQ]"
???

Figure 4: units in the caption should NOT be italic. Is the IVT the absolute value in any direction?

P6, L6: please also refer to Arnault et al., (2016) who have developed a similar WRF tracing tool, and, if relevant, specify the differences if there are any.

Figure 5: Does the northern boundary of the red zone correspond to the northern extent of the tropics (the Tropic of Cancer)? If not, why? If yes, please specify this. Moreover, why does the red zone in Domain CS2 have a corner. And why do the domains have the crypted names CS1 and CS2?

P7, L1: "*u* and *v* represent the wind field"
There is only **u** in the formulas…

Equations (1)-(4): Acronyms should never be in italic as this by convention means e.g., $I \cdot V \cdot T$ which is clearly different from IVT. The "d" of the integral should be roman as well. It is not clear what "*sfc*" means. The "mixing ratio *w*", whatever it may be, is somehow only dependent on *q*, then what is its function and how should this be interpreted? I do not understand this.

P7, L8: "water vapor mixing ratio (a), and tracer water vapor mixing ratio (b)"
What are these? Do they actually represent *q* and *w*? If so, why are their names suddenly different?

P8: What is WCB? Again, I suggest the authors to moderate their use of acronyms.

P8: Some inline equations are not according to conventions regarding the use of roman and italic fonts in physics. The symbol $\varphi$ is not being defined. Why is *Prec* not simply *P*? And why is "100" included in the equations? The formulas should be without 100 as they are fractions. When fractions are represented as percentages it is already implied (by calling them a percentage) that these are multiplied by 100% (not by 100 unitless).

P8, L27: "precipitation exceeded 3mm"
Per day? Per year? Per microsecond?

P8, L34-L35: "In the figure, there is evidence that the maximum of tropical moisture does not necessarily coincide with the low-level jet (LLJ), which is the maximum in wind speed at lower levels."
I am not a LJJ expert, but judging from the figure I do not clearly see that the place of the label is very different from the maximum of the integrated water vapor. Please elaborate or make the difference clearer in the figure.

Figure 6 and 7: these are certainly fascinating, probably more so when the reader would somehow be able to explore these interactively in 3D, but I am not so sure about the information content the way they are represented now. The white, blue and green colors of land and ocean are confusing with the other colors representing the water vapor mixing ratio and the cross sections blackish colors are not defined at all. Another point is that it seems that panels a-d do not correspond one-to-one with the rectangles provided at the bottom of each figure. My biggest problem is, however, with the terminology: Total water vapor mixing ratio provided in g/kg. It is not defined anywhere, but it seems this is just specific humidity (than call it specific humidity!). In any case a 'ratio' should always be unitless. Exactly the same comment applies to the tracers water vapor mixing ratio.

P9, L1: there is no Figure 8d

Figure 8: I guess this simply means the ratio of tagged water vapor to total water vapor (unitless!), but the caption provides the very cryptic description of "Tracers ratio in mixing ratio (g/kg)". There is no need to be so cryptic.

Figure 9: Here we can see some fascinating results as the tagged precipitation ratio is very different over the Iberian Peninsula. The higher elevations (Pyrenees, but also Galicia) have much higher tagged precipitation values compared to other areas. Beyond the Pyrenees in France to values have actually increased which seems counterintuitive. Does this have to do with the vertical distribution of tagged water and the rainfall generating processes which are not drawing the water specific-humidity weighted over the entire vertical column? Or does it have to do with the moment that precipitation falls? It would be great of the authors could elaborate on this.

Figure 10: How exactly is the position of the low level jet estimated?

P12, L6: "100mm"
It is daily precipitation, but it should still be mentioned whether is mm/day as one could also express daily precipitation in other units.

Figure 11: Why does the vertical axis show negative values? As mentioned before precipitation is a flux, thus cannot be expressed in mm. Please do not use computer code like "Tracers_prec", or "26_00h". An interesting question that can be raised here is whether the lack of tagged water during the initial rainfall is physical or whether it depends on the moment the simulation has started. In other words: when it started earlier, would the ratio of tagged water be apparent also during the initial precipitation?

P13, L13-L14: "It is well known that in a mature system, the water vapor store tends to be constant (e.g. Bullock and Johnson, 1971), and since the fate of tropical moisture is to precipitate sooner o**r** later, local convergence should keep the balance by lateral inflow."
What is a mature system? I cannot imagine that water vapor is constant during an extreme rainfall event. Please remove this statement or explain. Bullock and Johnson, 1971 is moreover missing from the reference list.

P14, L4: "behind or in front of the LJJ"
The maximum of tropical moisture being situated below the LJJ is seen in the figures, but where is it shown to be behind or in front of the LJJ.?

Figure A2: The LJJ estimation is missing here. Please also provide the correct unit for the latitudes in the caption.

Figure A3: I do not think this figure is referred to anywhere in the text.

**Technical corrections**
P1, L2: "3D Tracer tool" → 3D tracer tool

P1, L3: "Pacific Basin" → Pacific Ocean

P1, L12: Guan and Waliser (2015) **have** estimate**d** that

P1, L19: "several times of the discharge" → several times the discharge

P2, L11: Guan and Waliser (2015) **have** developed

P4, L7: "Model" → model

**References**

Arnault, J., Knoche, R., Wei, J. and Kunstmann, H.: Evaporation tagging and atmospheric water budget analysis with WRF: A regional precipitation recycling study for West Africa, Water Resour. Res., 52(3), 1544–1567, doi:10.1002/2015WR017704, 2016.

---

## Referee Comment (RC2) · B.R. Pagán (Referee) · 13 Jul 2017

General Comments

Eiras-Barca et al., develop a 3D tracer tool to track moisture origins for atmospheric river events. The tool itself is novel, although the findings are limited to two extreme events. The paper is well written however, there are a number of errors in the composition. Acronyms are defined repeatedly and are interchangeably used with full versions. There are also grammatical errors, which I have highlighted in the supplement. Some figures can be combined or entirely eliminated.

[Figure]

Regarding the content, with some clarifications the paper will be suitable for publication in ESD. Two points of concern are 1) the impact of excluding water vapor from the nudging scheme and 2) the discrepancy between WRF and observational precipitation in mountainous regions. It would be useful if the authors include a few lines regarding the local antecedent conditions for both events (i.e. soil moisture, snowpack) and the potential influence of these conditions on the partition of tropic versus local moisture originations. With two case studies, it is clear that this manuscript is more than just a methods paper for the new tracer tool. However, more emphasis should be placed on the novelty and distinction of this tool in comparison to other approaches like FLEXPART. Lastly, the conclusions should be stated more precisely.

I would encourage the authors to submit a follow-up paper once the tool has been widely implemented. It would be a valuable contribution to the scientific community to better understand how variations in moisture origins may impact AR events and vice versa. Some examples for comparisons, landfalling vs. non-landfalling, role of antecedent conditions, extreme vs. non-extreme events etc.

Specific Comments

Page 1, Line 13: "mean water vapor transport (IVT) of" Mean integrated water vapor transport.

Page 1, Line 17-18: "Between 3 and 5 ARs can be found per hemisphere at any given time," The 3-5 ARs in each hemisphere at any given time statistic is from Zhu and Newell 1998 not Guan and Waliser 2015.

Page 2, Line 16: Tropical moisture exports acronym defined as TME, but not used again in the text.

Page 2, Line 21: "Ramos et al. (2016) used the FLEXible PARTicle dispersion model (FLEXPART) to show that both tropical and local sources of moisture are present in AR landfall events for different European latitudes." Can the authors provide a better

distinction between the advantages/disadvantages of the Ramos et al 2016 Lagrangian tool versus the newly presented WRF-Tracer tool?

Page 3, Figure 1: "Source Era-In", use proper reference to ERA-Iterim. It is not intuitive which event is associated with the names "Great Coast Gale" and "Great Storm". I would stick with Pacific and Atlantic.

Page 3, Line 4: "This manuscript is organized as follows. Section 2 describes the applied data and methods, the results and discussion are presented in section 3 and we summarize our conclusions in section 4." Unnecessary description of paper outline.

Page 3, Line 7: Interchangeably using United States and U.S.

Page 3, Line: 11: How does Figure 1.a. (a snapshot in time) demonstrate the rapid development?

Page 4, Figure 2: All panels of this figure are repeated elsewhere in the manuscript, it should be removed.

Page 4, Line 1: "Regarding the alleged role of the atmospheric river in the fast deepening of the cyclone -35 mb in 24 hours- (Figure 1, b), Shutts (1990) showed the key role played by the latent heat release in the storm formation." Sentence is hard to follow, try rewording.

Page 4, Lines 7-9: "For the Pacific case, the WRF horizontal resolution is 15 km and the vertical column is divided into 40 levels. For the Atlantic simulation, grid spacing is 20 km in the horizontal and there are 50 vertical levels." Why use different resolutions?

Page 4, Line 10: Water Vapor Tracer (WVT) tool defined with acronym, but not used again.

Page 4, Lines 11-19: YSU, WSMC6, RRTM, ECMWF and ERA all undefined acronyms.

Page 4, Line 7, 14: "We use the Weather Research and Forecasting Model " "Spectral nudging has been applied" Make sure to use consistent verb tenses, present and past

tense are used interchangeably.

Page 4, Line 21: "Water vapor is not nudged, and given that the subject of this study is moisture transport and precipitation, we focus validations on these two variables" This sentence dismisses the nudging of water vapor since moisture transport and precipitation are used for validation. The statement makes it seem as though precipitation and moisture transport are not functions of water vapor. This should be further clarified and supported.

Page 5, Figure 3: Can you provide a difference map between the WRF simulation and observations for both locations? Not required in the text, but for this review.

Page 5, Line 5: "[FigVALQ]," Figure 4?

Page 5, line 8-9 and Figure 3: The overestimation of precipitation for the west coast event which is pronounced over high topography is concerning. Especially as the focus of this paper is moisture sources and differentiating between topic/subtropic and local origin. This is not to say that the observations are entirely accurate but do you have any supporting information to better clarify the amplified orographic enhancement? How will this potentially effect results?

Page 6, Figure 4: Same as Figure 3, provide a difference map.

Page 7, Figure 5: What do the labels of "Domain CS1"and "Domain CS2" mean? Again, avoid identifying the events by vague names of Great Coast Gale and Great Storm, use Pacific and Atlantic.

Page 7, Equation 1: Meridionial component of IVT? Incorrect formulation.

Page 7, Lines 8-9: "Figure 6 shows the three-dimensional distribution of water vapor mixing ratio (a), and tracer water vapor mixing ratio (b) for the event in the Pacific that made landfall along the U.S. West Coast on December 3, 2007." Date of landfall already mentioned in methods. Also West Coast alternates between being capitalized and not capitalized throughout the text.

Page 8, Line 12: "The main goal of Figures 6 and 7 is the visual depiction of the total and tracer moisture." Should not have to state this.

Page 8, Lines 13-21: This paragraph explaining Figure 1 should be moved to the methodology section where the figure was originally introduced and detailed.

Page 8, Line 22: The formatting of the names for the two events should be consistent, keeping it as the Pacific and Atlantic events is detailed enough. The inclusion of the dates is unnecessary.

Page 8, Line 30: "In the October 1987 Atlantic case, we also see a clear plume where tropical water vapor accounts for more than 80What is the explanation for the cause of rapid decrease?

Page 8, Line 34-35: "...there is evidence that the maximum of tropical moisture does not necessarily coincide with the low-level jet (LLJ), which is the maximum in wind speed at lower levels." Citation?

Page 9, Figure 6: The addition of lat/lon labels would make the figure and the point made on Page 8 line 5 more obvious. Also "d) Vertical cross sections of (d)." should be "sections of (b)"

Page 9, Line 1: Where is Figure 8d? Perhaps you meant 10d.

Page 11: Consider combining Figures 8 and 9.

Page 12, Figure 10: Define TCS. Axis labels of km cut off.

Page 12, Line 4: ""the Great Coast Gale of December, 2007"" Date not previously included in the quotes.

Page 13, Lines 5-7: "The Pacific event shows a more intense connection with tropical regions; therefore, the percentage of tropical precipitation for this event is higher and peaks at around 85These two main conclusions should be reworded.

Page 13, Line 8: "in terms of heavy precipitation" In terms of? Or chosen because of the subsequent heavy precipitation.

Page 14, Lines 7-9: "It is widely accepted in the literature that the bulk of moisture in ARs is primarily advected within the LLJ of extratropical cyclones but in light of our results we suggest that further discussion is necessary for this matter." This is not a very effective concluding sentence, should be reworded.

Page 16, Figure A2: Labels of LLJ missing. European not capitalized, however this event was not previously described as the "European case".

Page 20, Line 4: Partial citation.

Technical Corrections

See supplement.

Please also note the supplement to this comment:
https://www.earth-syst-dynam-discuss.net/esd-2017-63/esd-2017-63-RC2-supplement.pdf

---

## Referee Comment (RC3) · L. Wang-Erlandsson (Referee) · 14 Jul 2017

**General comments**

This manuscript introduces a new 3D Eulerian moisture tracking model and uses it in a WRF to analyse the moisture sources of two atmospheric river events. I interpret the goals of the paper as: (1) "to evaluate the relative contribution of tropical moisture to total water vapor and precipitation" in two AR events, (2) obtain information about the "vertical distribution of tropical moisture", (3) "as well as the position of the maximum of moisture with regard to the low-level jet". The study makes a contribution to the understanding of sources of atmospheric rivers and I think it is of interest for the ESD readership. However, there are ample opportunities to improve the presentation. Because of the large amount of minor errors and inconsistences (e.g., concerning units, colours, definitions, figure labels, acronyms, event name, etc.), I would suggest the authors to carefully proofread the manuscript before re-submission and not just check off the issues currently identified by the referees. I agree with the comments made by previous referees. Below are some general comments, specific comments, and technical corrections on issues that have not already been raised.

- **Introduction**
    - **Text editing.** The introduction comes across as unfocused. It is often difficult for the reader to extract the authors' key messages, understand how the provided information relates to the study, or fit in the context. For example, P1.L16 states that the "relationship between ARs and cyclones has been documented to be complex" without further explanation – what does that mean? P1.L22 list an enormous amount of references, but what do they say that is meaningful and relevant to mention in this context? In paragraph P2.L16, the importance of tropical moisture is first presented as a fact, just to be immediately contested by a list of findings from other studies. This leaves the reader wondering: Is there a real controversy? Is there a side that is more convincing? Why? What is this paragraph trying to convey? These are just examples of why I felt the introduction needs more editing. Please consider guiding the readers rather than just list information. Please also consider that in an efficient writing style, the first sentence of a paragraph ideally conveys the paragraph's key message.
    - **Add models overview**. Given that this paper introduces a new model, I would suggest the authors to add a paragraph that gives an overview of moisture tracking models, addressing questions such as: What are the main families of moisture tracking models (e.g., difference between Langrarian and Eulerian models)? What are their key differences in terms of tracking ARs? Do any of today's moisture tracking models trace moisture from a 3D volume? Is this study's research goal of investigating "vertical distribution of tropical moisture" something other models are able to do as well, and if so, how do they do it? Such an addition would put the new tracking model presented in this study in a context, allow the authors to better explain how their model differ from others, and help the reader understand what underpins the aims of this study. In the paragraph starting from P2.L16, there are also several references to moisture tracking models and methods that would become easier to understand if preceded by a model overview paragraph.
    - **Add paragraph on AR and LLJ**. In the conclusion, the authors state that "It is widely accepted in the literature that the bulk of moisture in ARs is primarily advected within the LLJ." However, this "acceptance" is not addressed in the Introduction. Given that relation to LLJ is one of the aims and findings of this study, I think the paper would benefit from having a dedicated paragraph reviewing the literature on

this topic. Such a review would also allow the authors to elaborate or discuss why they think their results differ from the literature.

- o **AR selection motivation**. Currently, the motivation for selecting the two AR events for the analyses are included in the methods. Perhaps a matter of taste, but I think it would flow better if the information of the two AR events is inserted before paragraph P2.L30, where the two events are first mentioned.
- **Methods**.
  - o What is the method used for AR detection? Is it one of the methods described in the introduction?
  - o How is the position of LLJ estimated?
- **Selection of AR events**. Why have the authors chosen to only analyse two events for which there are no possibility for comparisons with previous studies? Would it not be useful to also analyse AR event(s) that others have analysed, so that the results can be compared and potential differences discussed? Especially since the authors are introducing a new model, and challenging previous findings (on e.g., how ARs relate to LLJ)? I am not suggesting that it is absolutely necessary to include an analysis of a previously studied AR event, but I miss the discussion and reasoning behind the choice of not doing so.
- **Limitations and uncertainties**. Could you also discuss limitations and uncertainties in a separate paragraph?

**Specific comments**

- **Figures 1, 6 and 7**. Please consider adding corresponding videos in the supplementary information to show the temporal development in addition to the snapshots.
- **P4.L9 "10 days"**. How is that motivated? How much can the results be affected by the cut-off at 10 days?
- **P6.L5-6:** What difference do you expect between only tracking evaporation from the surface versus the presented approach of volume tracking? Could you discuss that?
- **P8.L1 "(sub)tropical"**: Tropical or subtropical? Which latitudes? Please specify what is considered "tropical moisture"?
- **P8.L4-10**: It is not clear how the information described can be interpreted from Figs. 6 and 7. E.g., how can the reader tell the relative tropical moisture content? How can the location of the "local convergence mechanism" be observed? Or how do we see that the tropical moisture contribution is less in Fig 7 than in Fig 6? Please consider providing clearer analyses and plots (e.g., of relative tropical contribution instead).
- **P8.L33-P9.L8 and Figure 10**: As LLJ is a key issue investigated, would it not be more informative to calculate and plot how often and when the maximum moisture actually coincides with the low level jet? Also, is LLJ simply defined as the "maximum in wind speed at lower levels" – or is there a wind speed threshold as well? If this is the definition, Fig 10b also shows maximum wind speeds spots at 1 km height, which coincide more with maximum moisture content – why are these spots not identified as LLJ?

**Technical corrections**

- **Figure 5.** What is the area highlighted in blue?
- **Figure 11.** Please label the vertical axes. Also, should the two axes scale really be identical? If not, consider changing the Ratio axis colour to red.
- **P7.Eq1**: Please define sfc.

---

## Author Comment (AC1) · 13 Sep 2017

**Supplemenatary Material : One-to-one answer to technical comments**

**P1, L2-L5: "the so-called "Great Coast Gale of 2007" in the Pacific Basin, and the "Great Storm of 1987" in the North Atlantic. Results show that between 80% and 90% of the moisture advected by the ARs, as well as between 70% and 80% of the associated precipitation have a tropical or subtropical origin."**

**I was intrigued by this statement and wondered about the different between atmospheric moisture and precipitation, however, it seems that the percentages are quite coarsely estimated from visual inspection (numbers for Canada would be much lower) and no hard conclusions can be drawn from this. I wonder whether more deterministic percentages could be calculated for when the AR event makes landfall or for precipitation occurring within x km from the coast during x days of the AR event. Moreover, the word subtropical does not come back anywhere in the paper. Why? Is it wrong in the abstract or in the rest of the paper?**

For the sake of clarity and following the recommendations of other reviewer as well, the term "sub-tropical" is going to be removed from the entire text. Regarding the reviewer's question about precipitation and humidity, we were referring to precipitation not just associated with the AR but with the entire system. In the cyclones we study, the maximum moisture amounts, in terms of integrated water vapor, are located along the AR; precipitation, however, is produced by moisture lifting mechanisms linked to the thermodynamic structure of the systems; thus it occurs not only in the cold frontal boundary, feeding on AR moisture, but also ahead of it, where air in the warm sector is forcibly ascending slantwise above the warm frontal zone, which is rather wide. We agree with the reviewer that the paragraph in the abstract is misleading, by making a one to one connection between AR and precipitation in the systems, and it should be rephrased.

With regard to the values of the tropical contribution to precipitation and to moisture content in the AR, they are indeed general coarse estimations by visual inspection. In the case of the ARs, the high percentage of tropical moisture above 80% is very clear; for precipitation, however, giving a single representative value for both regions as a whole is not so straightforward. We do not think that it is possible either to give a precise relation between distance from the coast and percentage of tropical contribution. As it is apparent in Fig 8, high values of this contribution are found as far inland as Wyoming or the Pyrenees, while lower values occur closer to shore. To address the reviewer's concern, we propose to correct the paragraph in the abstract as follows:

*"Results show that between 80% and 90% of the moisture advected by the ARs, and a high percentage of the total precipitation produced by the systems, have a tropical origin. The tropical contribution to precipitation is in general above 50%, and largely exceeds this value in the most affected areas."*

**P1, L8: it makes no sense at all to abbreviate 'mean water vapor transport' with IVT, moreover, the authors are not very consequent as on page 2 it suddenly appears that IVT stands for integrated water vapor flux. Judging from the acronym a logical term would be 'integrated vapor transport' or the acronym should be changed.**

It is our intent to be consistent in the use of the acronyms. Thus, the terms "mean water vapor transport" or "integrated water vapor flux" are going to be replaced by "Integrated Vapor Transport".

**P2, L17: " … West Coast" Reference?**

The following article is going to be used as a reference:

Dettinger, M., Ralph, F. M., & Lavers, D. (2015). Setting the stage for a global science of atmospheric rivers. *Eos*, *96*.

**P2: "By calculating the water vapor budget of 200 extratropical cyclones, Dacre et al. (2014) concluded that tropical moisture reaching the extratopics is only contributing to mid-level moisture, above the boundary layer." I think the authors find something else, it would be nice if they could reflect on this statement in their conclusions.**

We agree with the reviewer. We will add the reviewer's suggested comment to the last paragraph of the conclusions:
"Finally, our findings suggest that the maximum of tropical moisture does not necessarily coincide with the low-level jet of the extratropical cyclone in neither case analyzed. Instead, this maximum is located in near surface levels at lower latitudes to gradually ascend in northern latitudes, but still remaining below 2 km, mostly within the boundary layer, in contrast with findings in other studies (Dacre et al. (2004). The maximum of tropical moisture may be situated below and toward the back, or ahead of the LLJ, which is located along the cold front.

**Figures in general: Use clear headings instead of tiny names in the figure corners. The figures that show both AR events would benefit from clear titles specifying which panels are referring to the "Great Coast Gale" and the "Great Storm" respectively.**

Following the advice of the reviewer, clearer headings have been added to the latest version of the figures.

**Figure 1: The source should be spelled out ERA-Interim instead of ERA-In. "SLP" appears only 2 times in the paper, thus no need to abbreviate this. A scale of the IVT vectors is missing.**

Figure 1 has been remade following the indications of the reviewer.

**P3, L1: "total water vapor"**
**Does the tool also track liquid water and ice and the associated phase transitions? Please elaborate on this and the consequences for the results if the assumption is that it only tracks vapor. On the other hand the authors should adjust their terminology if the tool in fact includes liquid water and ice.**

Certainly, the tool also tracks liquid water and ice and the associated phase transitions. The final version of the manuscript is going to include an explicit reference to this. We will correct all instances where total water vapor should be referring to total moisture.

**P3, L3: "data and methods"**
**Then call section 2 data and methods instead of just methods**

Following the indications of another reviewer, the entire paragraph in the introduction section referring to the outline of the paper has been removed from the latest version of the manuscript. However, to be consistent with what we had before, as the reviewer points out, section 2 has been renamed "Data and Methods".

**P3, L3: "summarize our conclusions"**
**After reading section 4 it appears that you give a summary and conclusions, which is different from summarizing conclusions.**

As mentioned above, following the indications of another reviewer, this complete paragraph has been removed from the latest version of the manuscript.

**P3, L16: "Iberian Peninsula"**
**Later you refer to damages on the British Isles, thus the AR appears to extend further than the Iberian Peninsula.**

"Iberian Peninsula" has been replaced by "Iberian Peninsula and British Isles" in the latest version of the manuscript.

**Section 2: subheadings would be helpful for readability**

Subheadings "2.1: Data" and "2.2: Methods" have been added.

**P4: "YSU", "WSMC6", "RRTM"**
**What do these acronyms mean? The way they appear now they are not helpful for readability.**

The meaning of the acronyms of the parametrizations has been included.

**P4, L20: "spectral nudging"**
**Can the authors be more specific, as there are many ways to apply spectral nudging.**

We will add the following, to specify the way we applied spectral nudging:

Spectral nudging of waves longer than 1000 km, with a relaxation timescale of one hour and above the boundary layer only, has been applied to avoid distortion of the large scale circulation within the regional model domain due to the interaction between the model's solution and the lateral boundary conditions (Miguez-Macho et al., 2004, 2005)

**Figure 3: What is (L m-2)? This would read as Luminous intensity per square meter according to the International System of Units, which I hardly think the authors mean. The total precipitation is shown on land only, be specific about this in the caption. LIVNEH and IBERIA02 do not have to be capitalized.**

L·m$^{-2}$ has been replaced by mm, and the other reviewer's considerations have been followed in the latest version of the manuscript.

**P5, L6: "[FigVALQ]"???**

This was a LaTeX-related typo that will be corrected in the latest version of the manuscript.

**Figure 4: units in the caption should NOT be italic. Is the IVT the absolute value in any direction?**

The italic typography has been removed from the caption and "IVT" has been replaced by "Absolute value of IVT".

**P6, L6: please also refer to Arnault et al., (2016) who have developed a similar WRF tracing tool, and, if relevant, specify the differences if there are any.**

Arnault et al. (2016) will be cited in the Introduction section, when briefly discussing the moisture tracers tool. We leave the in-depth details of the method for another publication in review in this same ESD special issue:

Insua-Costa, D. and Miguez-Macho, G.: A new moisture tagging capability in the Weather Research and Forecasting Model: formulation, validation and application to the 2014 Great Lake-effect snowstorm, Earth Syst. Dynam. Discuss., https://doi.org/10.5194/esd-2017-80, in review, 2017

that we now cite in the text as well. The theory of the tracer method is the same in Arnault et al (2016) and our case; however we are not sure about the details of their implementation into WRF. The most evident difference is that Arnault et al (2016) does not include the tracers in the cumulus parametrization, which limits the

applicability of their tracers tool to high resolution simulations where convection can be assumed resolved Additionally, our tool is well validated and shows an error in traceability of much less than 1% (see the reference above).

**Figure 5: Does the northern boundary of the red zone correspond to the northern extent of the tropics (the Tropic of Cancer)? If not, why? If yes, please specify this. Moreover, why does the red zone in Domain CS2 have a corner. And why do the domains have the crypted names CS1 and CS2?**

The northern border of the red zone does not coincide with the Tropic of Cancer or any other parallel, because the domain is in a Lambert Conformal projection, and we defined the mask's border simply as a straight line in its native grid. The northern border of the mask approximately follows the Tropic of Cancer in the Pacific case, but it deviates some degrees to the north of it in the eastern part of the domain in the Atlantic case. We also left out of the mask, the small continental area of Africa included in the simulation's grid. This is because we initially performed experiments considering maritime evaporation only,,,to later on create the 3D masks by extending the original 2D masks in the vertical. This should have been corrected in the final experiments; however, since in the considered case, almost the entire moisture export from the tropics occurs in the western Atlantic, we do not think that these problems in experiment configuration alter our conclusions in any noticeable way.
We will change the crypted names CS1 and CS2 to Pacific and Atlantic, respectively.

**P7, L1: "u and v represent the wind field"**
**There is only u in the formulas…**

u in the formulas refers to the full wind vector. The following equation has been added for clarity:

$\mathbf{u} = (u,v)$

**Equations (1)-(4): Acronyms should never be in italic as this by convention means e.g., I V T** □ □which i **clearly different from IVT. The "d" of the integral should be roman as well. It is not clear what "sfc" means. The "mixing ratio w", whatever it may be, is somehow only dependent on q, then what is its function and how should this be interpreted? I do not understand this.**

All the suggested changes about conventions and the clarification of "sfc" (surface) will be followed. The considerations regarding the relationship between the specific humidity (q) and the mixing ratio (w) are formally necessary. By definition they are not the same variable. Mixing ratio is the ratio of moisture mass to mass of dry air, whereas specific humidity is the ratio of moisture mass to total air mass. Because moisture mass is only a very small fraction of total air mass, they are approximately equal. The point is that the extendedly used variable is "q", but WRF operates with "w", and with eq 4 we just show that we assume the approximation q=w .

**P7, L8: "water vapor mixing ratio (a), and tracer water vapor mixing ratio (b)" What are these? Do they actually represent q and w? If so, why are their names suddenly different?**

"w" is the accepted acronym for the water vapor mixing ratio. As we explained above, the direct output from the model is in terms of mixing ratios. But according to eq 4, we assume that specific humidity (q) and mixing ratio (w) are totally interchangeable. For the sake of simplicity, all this terminology has been reviewed in the latest version of the manuscript.

**P8: What is WCB? Again, I suggest the authors to moderate their use of acronyms.**

P1L14 of the original version of the manuscript reads "ARs transport moisture towards the Warm Conveyor Belts (WCBs) of extratropical cyclones by thermal wind advection". It is a commonly used acronym in dynamic meteorology, but in consideration to a broader audience, it has been replaced by the full name in the final version of the manuscript.

**P8: Some inline equations are not according to conventions regarding the use of roman and italic fonts in physics. The symbol φ is not being defined. Why is Prec not simply P? And why is "100" included in the equations? The formulas should be without 100 as they are fractions. When fractions are represented as percentages it is already implied (by calling them a percentage) that these are multiplied by 100% (not by 100 unitless).**

"100"s are going to be removed from the latest version of the manuscript, and φ is going to be defined as "the latitude of the point". However, since concepts like "explosive cyclogenesis", "cyclones", etc. are on the table; we think that the use of P should be avoided for precipitation in order to not be confused with "pressure".

**P8, L27: "precipitation exceeded 3mm" Per day? Per year? Per microsecond?**

"per day" is going to be added to the final version of the manuscript.

**P8, L34-L35: "In the figure, there is evidence that the maximum of tropical moisture does not necessarily coincide with the low-level jet (LLJ), which is the maximum in wind speed at lower levels."**
**I am not a LJJ expert, but judging from the figure I do not clearly see that the place of the label is very different from the maximum of the integrated water vapor. Please elaborate or make the difference clearer in the figure.**

Following the instructions of all the reviewers, a more detailed discussion on this matter is going to be included in the final version of the manuscript.

**Figure 6 and 7: these are certainly fascinating, probably more so when the reader would somehow be able to explore these interactively in 3D, but I am not so sure about the information content the way they are represented now. The white, blue and green colors of land and ocean are confusing with the other colors representing the water vapor mixing ratio and the cross sections blackish colors are not defined at all. Another point is that it seems that panels a-d do not correspond one-to-one with the rectangles provided at the bottom of each figure. My biggest problem is, however, with the terminology: Total water vapor mixing ratio provided in g/kg. It is not defined anywhere, but it seems this is just specific humidity (than call it specific humidity!). In any case a 'ratio' should always be unitless. Exactly the same comment applies to the tracers water vapor mixing ratio.**

As mentioned above, "mixing ratio" is defined as the ratio of mass of water vapor to mass of **dry air,** while specific humidity is defined as the ratio of mass of water vapor to mass of **wet air** (dry air+water vapor). Since the absolute amount of water vapor in the air is very small, the difference between both variables is negligible, and we assume they are totally interchangeable. Mixing ratio is defined in eq (4).

Regarding the question of the units, both specific humidity and mixing ratio are ratios, thus unitless. However, it is usually more readable and convenient working with g/kg rather than kg/kg. Therefoe, even when it might not be formally proper, it is very common in the literature to use g/kg as units for both mixing ratios and specific humidity, and we would prefer to keep it this way.

**P9, L1: there is no Figure 8d**

The typo has been corrected in the latest version of the manuscript. We meant Figure 10.d

**Figure 8: I guess this simply means the ratio of tagged water vapor to total water vapor (unitless!), but the caption provides the very cryptic description of "Tracers ratio in mixing ratio (g/kg)". There is no need to be so cryptic.**

The tags of the information displayed in Figure 8 has been reviewed and simplified in the latest version of the manuscript following the reviewer's suggestion.

**Figure 9: Here we can see some fascinating results as the tagged precipitation ratio is very different over the Iberian Peninsula. The higher elevations (Pyrenees, but also Galicia) have much higher tagged precipitation values compared to other areas. Beyond the Pyrenees in France to values have actually increased which seems counterintuitive. Does this have to do with the vertical distribution of tagged water and the rainfall generating processes which are not drawing the water specific-humidity weighted over the entire vertical column? Or does it have to do with the moment that precipitation falls? It would be great of the authors could elaborate on this.**

The reviewer is right in that the pattern of tropical contribution to precipitation is likely linked to the different dynamic mechanisms of precipitation genesis in the system. In addition the blocking effect of the land mass and mountain ranges of the Iberian Peninsula and North America is also an important factor. The high percentage of tagged precipitation in mountain ranges such as the Pyrenees and the Rockies, far ahead of the cold front and the AR associated with the systems, is related to moisture transport and slantwise lift in mid-levels along the warm frontal boundary. As it is apparent in Fig 8, this moisture is mostly of tropical origin in both cases. We will include a discussion on this in the final version of the manuscript.

**Figure 10: How exactly is the position of the low level jet estimated?**

The position of the level jet is estimated as the local maximum in the wind speed in low levels. Following the instructions of all the reviewers, a further discussion on this matter is going to be included in the final version of the manuscript.

**P12, L6: "100mm"**
**It is daily precipitation, but it should still be mentioned whether is mm/day as one could also express daily precipitation in other units.**

"mm" has been replaced by "mm/day" in the last version of the manuscript.

**Figure 11: Why does the vertical axis show negative values? As mentioned before precipitation is a flux, thus cannot be expressed in mm. Please do not use computer code like "Tracers_prec", or "26_00h". An interesting question that can be raised here is whether the lack of tagged water during the initial rainfall is physical or whether it depends on the moment the simulation has started. In other words: when it started earlier, would the ratio of tagged water be apparent also during the initial precipitation?**

Negative values in the ratios are due to computational noise occurring in areas with negligible amounts of precipitation. This values are going to be removed from the axis. When we refer to precipitation we do not mean the flux of water in a parcel of air, but to the total amount of water accumulated in a square meter at the surface. This is usually measured in $kg/m^2$ or $l/m^2$, which are commonly converted to equivalent height of water in "mm".
With regard to the timing of tagged precipitation, the evolution of the system in Fig. 1 shows that the air mass ahead of the system, just west of the North American coast, had very likely very little tropical moisture content or connection, since winds were from the north and IWV values were relatively low. So, we do not think that starting the simulation earlier would result in more tropical contribution in the initial precipitation, ahead of the cold front.

**P13, L13-L14: "It is well known that in a mature system, the water vapor store tends to be constant (e.g. Bullock and Johnson, 1971), and since the fate of tropical moisture is to precipitate sooner or later, local convergence should keep the balance by lateral inflow." What is a mature system? I cannot imagine that**

**water vapor is constant during an extreme rainfall event. Please remove this statement or explain. Bullock and Johnson, 1971 is moreover missing from the reference list.**

In meteorology, a mature system is usually considered when the baroclinic structures (fronts) are well formed and differentiated.

The water vapor during extreme rainfall is certainly not constant, or course. What the reference paper and we mean, is that the amount of water vapor tends to be constant in the system as a whole, that is, in the system, the gains of moisture from convergence approximately compensate the losses due to precipitation. Strong systems with high precipitation rates tend to also have robust convergence, which results in balanced moisture contents. The sentence is going to be properly clarified and the citation included in the reference list.

**P14, L4: "behind or in front of the LJJ"**
**The maximum of tropical moisture being situated below the LJJ is seen in the figures, but where is it shown to be behind or in front of the LJJ.?**

Following the instructions of all the reviewers, a more detailed discussion on this matter is going to be included in the final version of the manuscript. The maximum below is very clear in several of the cross sections shown. The maximum ahead is only apparent in Fig. 10c and by behind, we really meant toward the back, when the maximum is below.

**Figure A2: The LJJ estimation is missing here. Please also provide the correct unit for the latitudes in the caption.**

The LLJ estimations and the units are going to be included in the final version of the manuscript.

**Figure A3: I do not think this figure is referred to anywhere in the text.**

This figure is going to be removed from the manuscript.

**Technical corrections**

**P1, L2: "3D Tracer tool" > 3D tracer tool**
**P1, L3: "Pacific Basin" > Pacific Ocean**
**P1, L12: Guan and Waliser (2015) have estimated that**
**P1, L19: "several times of the discharge" > several times the discharge**
**P2, L11: Guan and Waliser (2015) have developed**
**P4, L7: "Model" > model**

All the technical corrections have been made in the latest version of the manuscript.

---

## Author Comment (AC2) · 14 Sep 2017

**Supplemenatary Material : One-to-one answer to technical comments**

**Page 1, Line 13: "mean water vapor transport (IVT) of" Mean integrated water vapor transport.**

"Mean water vapor transport (IVT)" will be replaced by "Mean integrated water vapor transport (IVT)" in the final version of the manuscript.

**Page 1, Line 17-18: "Between 3 and 5 ARs can be found per hemisphere at any given time," The 3-5 ARs in each hemisphere at any given time statistic is from Zhu and Newell 1998 not Guan and Waliser 2015.**

The reviewer is correct in that the estimation of "between 3 and 5 ARs" was initially proposed by Zhu and Newell (1998), whereas Guan and Waliser (2015) provide a more up-to-date information about the total amount of meridional moisture transport by ARs per hemisphere. We will include both references in the final version of the manuscript:

*Between 3 and 5 ARs can be found per hemisphere at any given time (Zhu and Newell, 1998), accounting for approximately 84% of the meridional IVT for the Northern Hemisphere and about 88% in the Southern Hemisphere (Guan and Waliser, 2015).*

**Page 2, Line 16: Tropical moisture exports acronym defined as TME, but not used again in the text.**

The acronym will be removed in the final version of the manuscript.

**Page 2, Line 21: "Ramos et al. (2016) used the FLEXible PARTicle dispersion model (FLEXPART) to show that both tropical and local sources of moisture are present in AR landfall events for different European latitudes." Can the authors provide a better distinction between the advantages/disadvantages of the Ramos et al 2016 Lagrangian tool versus the newly presented WRF-Tracer tool?**

It was not our intent to present the WRF-Tracer tool in this paper, contrasting it with other methods. Our goal was to just use it to answer specific questions about the importance of tropical moisture sources in ARs. We will make this point more explicit, as we leave the in-depth details of the method and the discussions of advantanges/disadvanges with respect to other methods for another publication in review in this same ESD special issue:

Insua-Costa, D. and Miguez-Macho, G.: A new moisture tagging capability in the Weather Research and Forecasting Model: formulation, validation and application to the 2014 Great Lake-effect snowstorm, Earth Syst. Dynam. Discuss., https://doi.org/10.5194/esd-2017-80, in review, 2017

that we will now cite in the text. However, at the reviewer's request, we will also include a brief discussion about the key differences between lagrangian models and online eulerian models, as our WRF-Tracer tool. Lagrangian models are based on the spatio-temporal tracking of individual fluid particles. They give information about the Evaporation-Precipitation budget and as they rely on atmospheric analyses, they need to estimate subgrid vertical mixing occurring in the column. Online euclearian models like our WRF-Tracer tool, explicitly calculate the evolution of the moisture from a given source, as they are coupled to an atmospheric model. The main disadvantage is that their accuracy is tied to the skill of the atmospheric model. If simulations are not realistic, moisture pathways will not be either.

**Page 3, Figure 1: "Source Era-In", use proper reference to ERA-Iterim. It is not intuitive which event is associated with the names "Great Coast Gale" and "Great Storm". I would stick with Pacific and Atlantic.**

The final version of the manuscript will include this caption for Figure 1:

*Figure 1. Integrated vapor transport (IVT, vectors, kg·m⁻¹·s⁻¹), sea level pressure (SLP, isobars, hPa) and integrated water vapor (IWV, background. kg·m⁻²) for both the Pacific "Great Coast Gale" (a-d) and the Atlantic "Great Storm" (e-h) avents throughout a four-days window time frame. Source: ERA-Interim (Barrisfor et al., 2009).*

**Page 3, Line 4: "This manuscript is organized as follows. Section 2 describes the applied data and methods, the results and discussion are presented in section 3 and we summarize our conclusions in section 4." Unnecessary description of paper outline.**

Following the reviewer's suggestion, the description of the paper outline is going to be removed from the introduction section.

**Page 3, Line 7: Interchangeably using United States and U.S.**

The acronym "U.S." is going to be replaced by "United States" throughout the entire text in the final version of the paper.

**Page 3, Line: 11: How does Figure 1.a. (a snapshot in time) demonstrate the rapid development?**

Certainly, a snapshot in time would never demonstrate a rapid development. This is going to be replaced by Figure 1a-d in the final version of the manuscript.

**Page 4, Figure 2: All panels of this figure are repeated elsewhere in the manuscript, it should be removed.**

Following your advice, we have decided to remove all the repeated figures from the final version of the manuscript**.**

**Page 4, Line 1: "Regarding the alleged role of the atmospheric river in the fast deepening of the cyclone -35 mb in 24 hours- (Figure 1, b), Shutts (1990) showed the key role played by the latent heat release in the storm formation." Sentence is hard to follow, try rewording.**

For the sake of clarity, the sentence is going to be completely reworded as follows:
For this case, Shutts (1990) showed that two thirds of the central pressure falling could be ascribed to latent heat release, which suggest that the existent atmospheric river played a key role in the fast deepening of the cyclone 35 hPa of pressure drop in 24 hours- (Figure 1, b).

**Page 4, Lines 7-9: "For the Pacific case, the WRF horizontal resolution is 15 km and the vertical column is divided into 40 levels. For the Atlantic simulation, grid spacing is 20 km in the horizontal and there are 50 vertical levels." Why use different resolutions?**

This study is the result of collaboration between our groups at USC and UIUC. The simulation carried out for the Atlantic case is part of a series of analyses that we are developing for Europe at the Non-Linear Physics Group at USC. We performed the simulation for the US West Coast as part of a different set of analyses that are being carried out at the Department of Atmospheric Sciences at the UIUC. The spatial resolution of the experiments had been forehand selected by both departments some years ago and is slightly different, since the domain sizes are also different. The domain corresponding to the Atlantic case is much bigger than the Pacific one; thus, a slight reduction of the resolution is well justified in order to keep the computational requirements of the simulations relatively low. We do not think that these differences have any significant effect in the conclusions of the present study.

**Page 4, Line 10: Water Vapor Tracer (WVT) tool defined with acronym, but not used again.**

The acronym is going to be removed from the final version of the manuscript.

**Page 4, Lines 11-19: YSU, WSMC6, RRTM, ECMWF and ERA all undefined acronyms.**

The definitions of the acronyms are going to be included in the final version of the manuscript.

**Page 4, Line 7, 14: "We use the Weather Research and Forecasting Model " "Spectral nudging has been applied" Make sure to use consistent verb tenses, present and past tense are used interchangeably.**

The present tense is going to be used throughout the entire text.

**Page 4, Line 21: "Water vapor is not nudged, and given that the subject of this study is moisture transport and precipitation, we focus validations on these two variables" This sentence dismisses the nudging of water vapor since moisture transport and precipitation are used for validation. The statement makes it seem as though precipitation and moisture transport are not functions of water vapor. This should be further clarified and supported.**

The reviewer is right in that including both statements in the same sentence may lead to confusion. Thus, in the new version of the manuscript, we propose to rephrase it as follows:

*Water vapor is not nudged to ensure the mass conservation needed for the traceability of humidity from different sources. Given that the subject of this study is moisture transport and precipitation, we focus validations on these two variables.*

**Page 5, Figure 3: Can you provide a difference map between the WRF simulation and observations for both locations? Not required in the text, but for this review.**

Pacific:

WRF Simulation of Precipitation – Livneh Precipitation :

[Figure]

Atlantic:

WRF Simulation of Precipitation – IBERIA02 Precipitation : Figure 4.

[Figure]

Regarding the last map, certainly the observed differences are high. We have detected that this is due to the fact that IBERIA time range cover from 12PM to 12PM, and we were comparing with 0AM-0AM WRF. With this correction , the validation improves substantially. This is going to be fixed in the final version of the manuscript. (See Figure):

[Figure]

**As mentioned in the text, the model tends to overestimate precipitation in the mountains, precisely where analyses tend to underestimate it, so the differences are largely magnified.**

**Page 5, Line 5: "[FigVALQ]," Figure 4?**

Certainly, this typo is going to be corrected in the final version of the manuscript.

**Page 5, line 8-9 and Figure 3: The overestimation of precipitation for the west coast event which is pronounced over high topography is concerning. Especially as the focus of this paper is moisture sources and differentiating between topic/subtropic and local origin. This is not to say that the observations are entirely accurate but do you have any supporting information to better clarify the amplified orographic enhancement? How will this potentially effect results?**

Certainly, model simulations in areas where orographic amplification could play a key role, still require substantial improvements. This lack of accuracy in abrupt orography areas has its bases in a insufficient resolution (present in all the mesoscale models), wich is not the proper to depict the mountains. This essentially causes that the vertical fluxes are not well solved.

However, we think that the important point in here is the ratio of tropical precipitation out of the total. Even when the total amount of precipitation could not be well solved, we do not find any reason to think that the ratio could have the same problem.

**Page 6, Figure 4: Same as Figure 3, provide a difference map.**

[Figure]

[Figure]

**Page 7, Figure 5: What do the labels of "Domain CS1"and "Domain CS2" mean? Again, avoid identifying the events by vague names of Great Coast Gale and Great Storm, use Pacific and Atlantic.**

Following the advice of the reviewer, all those labels and vague names are going to be removed or renamed in the final version of the manuscript.

**Page 7, Equation 1: Meridonial component of IVT? Incorrect formulation.**

Actually, **u** is a vector magnitude representing **u**=(u,v). Since the template of the review forces us to use bold letters and not arrows to represent vectors, no changes can be made in the equation. The following equation has been added for clarity:
**u** = (u,v)

**Page 7, Lines 8-9: "Figure 6 shows the three-dimensional distribution of water vapor mixing ratio (a), and tracer water vapor mixing ratio (b) for the event in the Pacific that made landfall along the U.S. West Coast on December 3, 2007." Date of landfall already mentioned in methods. Also West Coast alternates between being capitalized and not capitalized throughout the text.**

The date is going to be removed and the use of capitalized words is going to be consistent throughout the text in the latest version of the manuscript.

**Page 8, Line 12: "The main goal of Figures 6 and 7 is the visual depiction of the total and tracer moisture." Should not have to state this.**

This statement is going to be removed from the final version of the manuscript.

**Page 8, Lines 13-21: This paragraph explaining Figure 1 should be moved to the methodology section where the figure was originally introduced and detailed.**

Following the instructions of the reviewer, the paragraph is going to be moved to the proper location.

**Page 8, Line 22: The formatting of the names for the two events should be consistent, keeping it as the Pacific and Atlantic events is detailed enough. The inclusion of the dates is unnecessary.**

The dates are going to be removed, and the names of the events are going to be consistently used in the final version of the manuscript.

**Page 8, Line 30: "In the October 1987 Atlantic case, we also see a clear plume where tropical water vapor accounts for more than 80%. What is the explanation for the cause of rapid decrease?**

At the time shown in Fig. 9, the center of the storm is just off the coast of Galicia (see Fig 2), which lies at the converging point between cold and warm front. Airflows are complex at that location within baroclinic systems, with air in the warm conveyor belt, loaded with tropical moisture in this case, lifted above air ahead of the warm front, with much less tropical content. In addition, enhanced convergence due to the explosive cyclogenesis taking place also feeds local moisture into the AR, decreasing the tropical moisture fraction in it. The presence of the high terrain of the Iberian Peninsula makes the situation harder to analyze.
. To address the reviewers concerns, we propose to reword the sentence as follows:

"In the October 1987 Atlantic case, we also see a clear plume where tropical water vapor accounts for more than 80% of precipitable water; however, the percentage decreases to around 70% closer to the center of the system and just before arriving on the Iberian coast (Figure 9). In this case, cyclogenesis occurs just off the coast of Galicia, on the northwest tip of the Iberian Peninsula, and thus, the enhanced convergence of existent local moisture feeds the AR and is involved in the heavy precipitation, which consequently is only between 60% and 80% of tropical origin.

**Page 8, Line 34-35: ": : :there is evidence that the maximum of tropical moisture does not necessarily coincide with the low-level jet (LLJ), which is the maximum in wind speed at lower levels." Citation?**

This is an evidence obtained from our results, so no citations are required. This is now going to be clarified in the text.

**Page 9, Figure 6: The addition of lat/lon labels would make the figure and the point made on Page 8 line 5 more obvious. Also "d) Vertical cross sections of (d)." should be "sections of (b)"**

The 3D plotting software is not specifically designed for earth sciences. The input data should be in x-y projection, so, adding lat/lon to labels is not yet an easy task in this version of the program. Instead, to address the reviewer's concern, we will add the lat/lon labels to the map box located below (c), indicating the area that is being plotted in 3D above. The typo "sections of (d)" is going to be replaced by "sections of (b)".

**Page 9, Line 1: Where is Figure 8d? Perhaps you meant 10d.**

Certainly, we meant 10d. This typo is going to be fixed in the next version of the manuscript.

**Page 11: Consider combining Figures 8 and 9.**

Following the reviewer's advice, Figures 8 and 9 are going to be combined in the next version of the manuscript.

**Page 12, Figure 10: Define TCS. Axis labels of km cut off.**

"TCS" is going to be replaced by "Transversal Cross Sections" and the axis labels will be fixed in the final version of the manuscript.

**Page 12, Line 4: ""the Great Coast Gale of December, 2007"" Date not previously included in the quotes.**

For consistency, quotes will only include "Great Coast Gale" anywhere in the text .

**Page 13, Lines 5-7: "The Pacific event shows a more intense connection with tropical regions; therefore, the percentage of tropical precipitation for this event is higher and peaks at around 85These two main conclusions should be reworded.**

Following the reviewer's advice, these conclusions are going to be reworded as follows:
The Pacific event shows a more intense connection with tropical regions than the Atlantic case. As a result, the percentage of tropical precipitation for this event over North America is higher and peaks at around 85%. Nevertheless, for the Atlantic event, still more than 60% of the resulting precipitation is of tropical origin

**Page 13, Line 8: "in terms of heavy precipitation" In terms of? Or chosen because of the subsequent heavy precipitation.**

"in terms of" is going to be replaced by *"chosen because of the subsequent"*

**Page 14, Lines 7-9: "It is widely accepted in the literature that the bulk of moisture in ARs is primarily advected within the LLJ of extratropical cyclones but in light of our results we suggest that further discussion is necessary for this matter." This is not a very effective concluding sentence, should be reworded.**

The sentence will be reworded as follows:

*"It is widely accepted in the literature that the bulk of moisture in ARs is primarily advected within the LLJ of extratropical cyclones but our results suggest that this is not always the case, and that a revision with a more in-depth investigation is necessary for this matter."*

**Page 16, Figure A2: Labels of LLJ missing. European not capitalized, however this event was not previously described as the "European case".**

The labels of the LLJ are going to be added, and "European case" is going to be replace by Atlantic case, as named elsewhere in the text,

**Page 20, Line 4: Partial citation.**

This typo is going to be fixed .

---

## Author Comment (AC3) · 26 Sep 2017

**Supplemenatary Material : One-to-one answer to technical comments**

**Comments Regarding the introduction:**

We thank the reviewer for the advices. The introduction is going to be rewritten taking into account all the reviewer's comments.

**Add models overview**

It was not our intent to present the WRF-Tracer tool in this paper. Our goal was to just use it to answer specific questions about the importance of tropical moisture sources in ARs. We will make this point more explicit, as we leave the in-depth details of the method and the discussions of advantanges/disadvanges with respect to other methods for another publication in review in this same ESD special issue:

Insua-Costa, D. and Miguez-Macho, G.: A new moisture tagging capability in the Weather Research and Forecasting Model: formulation, validation and application to the 2014 Great Lake-effect snowstorm, Earth Syst. Dynam. Discuss., https://doi.org/10.5194/esd-2017-80, in review, 2017

that we will now cite in the text. However, as per the reviewer's suggestion, we will also include a brief discussion stating the main characteristics and the advantages of using our tracer tool.

**Add paragraph on AR and LLJ**

One of the motivations of the paper was to contribute to the on-going discussion in the community to provide a definition of Atmospheric River for the Glossary of Meteorology. The current draft definition reads as follows:

"A long narrow and transient corridor of anomalously strong horizontal water vapor transport that is typically located in the lowest 3 km of the troposphere and associated with a low-level jet stream ahead of the cold front of an extratropical cyclone. The water vapor in atmospheric rivers is supplied by tropical and/or extratropical moisture sources and atmospheric rivers frequently lead to heavy precipitation where they intersect topographic or other lower-tropospheric boundaries, or enter into the warm-conveyor-belt-related isentropic upward air motion. Atmospheric rivers conduct over 90% of all poleward water vapor transport in the extratropics in less than 10% of the zonal circumference of the globe.

(Please, see https://annual.ametsoc.org/2017/index.cfm/programs/town-hall-meetings/atmospheric-rivers-a-discussion-of-the-definition-under-development-for-the-glossary-of-meteorology/)

The low level jet is mentioned in the first sentence of the proposed definition, implicitly equaling it to the AR itself. Furthermore, in this same sentence, ARs are defined purely in terms of moisture transport, and not in terms of moisture content, hence giving further weight to the low level jet concept. Most detection algorithms in the literature, however, use both water vapor flux and water vapor content to define ARs, to avoid strong jets with average moisture loads to be confused with ARs. We tend to favor a characterization of ARs that includes both moisture transport and moisture content, and thus, with the statement the reviewer is referring to ("It is widely accepted in the literature that the bulk of moisture in ARs is primarily advected within the LLJ.") we were, perhaps not very effectively, transmitting that we do not agree with identifying low level jet and AR.

Our results do not conflict with the existent literature, especially with seminal observationally based studies like Ralph et al. (2004, 2005). We find that most of the moisture content AND transport defining ARs occur in the lower 3km and in a narrow zone along the cold front. This is where the LLJ is located, and this maximum in wind is also a local maximum in moisture transport. Our point is that moisture content (IWV) is also a big part of what defines an AR, and, as we show in the cases we study, the bulk of the moisture load is clearly not associated with the LLJ. Most moisture transport does occur in the lower 3km, as the proposed definition states, but not just within the LLJ.

As per the reviewer's request, a paragraph discussing the relationship between the AR and the LLJ is going to be added to the introduction. In particular we will refer to observational studies like Ralph et al (2004, and 2005), already cited in the paper, that describe the vertical structure of moisture and moisture transport in ARs. We will also rephrase the concluding paragraph to make our point more clear and to avoid confusion

**Add selection motivation**

The motivation in the selection of the events is going to be expanded (see below)

**Comments regarding  Methods:**

**o What is the method used for AR detection? Is it one of the methods described in the introduction?**

Both cases are some of the most intense, documented and discussed AR-events over the Atlantic and Pacific basins, although much of these discussions refer to impacts, rather than to the structure of the ARs themselves. These events are very well detected by all methods and algorithms published in the literature (GUAN2015, BRANDS2016 and EIRAS2016). The lack of agreement among different algorithms in the detection of ARs usually occurs in much weaker cases. Following the advice of the reviewer we are going to add this sentence to the final version of the manuscript:          *"... both events are very intense in terms of IVT and IWV, and are well detected by different detection methods -GUAN2015, EIRAS2016 and BRANDS2016-."*

**o How is the position of LLJ estimated?**

The position of the LLJ is estimated as a relative maximum in the module of the wind at lower levels, around 1km elevation, just ahead of the cold frontal boundary..

**• Selection of AR events. Why have the authors chosen to only analyse two events for which there are no possibility for comparisons with previous studies? Would it not be useful to also analyse AR event(s) that others have analysed, so that the results can be compared and potential differences discussed? Especially since the authors are introducing a new model, and challenging previous findings (on e.g., how ARs relate to LLJ)? I am not suggesting that it is absolutely necessary to include an analysis of a previously studied AR event, but I miss the discussion and reasoning behind the choice of not doing so.**

We selected the events in terms of socio-economic impacts (extreme precipitation, flooding and damages), and because they are very intense paradigmatic Atmospheric River cases, especially the Pacific case. It is true however, that there are no other studies on the thermodynamic structure of these systems, focusing explicitly on the associated Atmospheric River and the origin of moisture within. So, a comparison like the suggested by the reviewer is not possible. Notwithstanding, we think that these two cases illustrate very well the two main points we wanted to make. First, that a high tropical moisture content is likely needed for Atmospheric Rivers to produce extreme precipitation events and second, that the bulk of moisture content that defines the Atmospheric River does not have to coincide with the low level jet. (see discussion above on LLJ and ARs).  The latter point is

somewhat more controversial, because in some cases ARs are defined purely on atmospheric moisture transport (like in the proposed definition mentioned previously), whereas in others (such as in detection algorithms) ARs are defined in terms of both atmospheric content and atmospheric transport.

We will add a paragraph discussing the motivations in the selection of the events when reforming the introduccion section, per the reviewer's suggestion.

• **Limitations and uncertainties. Could you also discuss limitations and uncertainties in a separate paragraph?**

The limitations and uncertainties of the tracers tool are those associated with the WRF simulations, as demonstrated in the Insua-Costa and Miguez-Macho (2017) paper about the method, that we will now cite. If we accept the results of WRF in terms of moisture transport, distribution, etc, as accurate, then the tracer tool running coupled to the model can separate moisture from different sources with a very small error (much less than 1% in traceability). Thus, the tracer tool is very accurate in the "model world", and the uncertainty in the "real word" is due to the WRF model error in simulating the systems. Following the advice of the reviewer, the limitations and uncertainties are going to be discussed in the final version of the manuscript, both in the introduction and the conclusions.

**Specific comments**

• **Figures 1, 6 and 7. Please consider adding corresponding videos in the supplementary information to show the temporal development in addition to the snapshots.**

Following the advice of the reviewer, we are going to edit and include the videos in the supplementary information of the manuscript.

• **P4.L9 "10 days". How is that motivated? How much can the results be affected by the cutoff at 10 days?**

The reason for performing a 10 day simulation is to capture the full complex development of both systems. As shown in Fig 1, for the Pacific case, there is a preexistent plume of tropical moisture in the precursor storm, already 5 days before making landfall. In the Atlantic case, most of the tropical moisture export occurs from the Caribbean area, even before those 5 days. Thus, in order to properly track moisture from its tropical source, the extended 10 day period is needed. Results would be significantly affected if using shorter periods because we would be missing a large portion of the tropical moisture actually involved in the storms. We will clarify this point when stating the length of the simulations.

• **P6.L5-6: What difference do you expect between only tracking evaporation from the surface versus the presented approach of volume tracking? Could you discuss that?**

A 3D source is needed in our experiments to make sure that the moisture in the tropical air mass intervening in the development of the simulated storms is properly tagged. To obtain similar results with a 2D source, we would have to, first, include the whole tropical region in our domains, and second, perform much longer simulations, to ensure that the moisture in the tropical air mass is entirely made up of water vapor evaporated within our simulated period and domain, and therefore, tagged.

We are not intending to contrast two different approaches, with 3D sources vs 2D sources, as those used in a previous study. We developed tracking from a 3D source specifically to address the question in this paper about the tropical contribution to moisture in ARs. To avoid confusion, we will rephrase the paragraph in P6, eliminating references to previous studies, as follows:

"The Eulerian tracer tool operates as follows: A wide region in the domain covering the tropical latitudes is set up as a three dimensional tracer mask. All the water vapor in this three-dimensional volume (including the water vapor evaporated and advected into the masked region) is tracked in space and time.,

**• P8.L1 "(sub)tropical": Tropical or subtropical? Which latitudes? Please specify what is considered "tropical moisture"?**

All areas inside the masks depicted in Fig 5 are considered tropical latitudes in the study. These correspond approximately to latitudes below the Tropic of Cancer, with some deviations due to the lambert conformal projection used in the model grid in which the masked regions are defined. We will remove mentions to the sub-tropics to avoid confusion.

**• P8.L4-10: It is not clear how the information described can be interpreted from Figs. 6 and 7. E.g., how can the reader tell the relative tropical moisture content? How can the location of the "local convergence mechanism" be observed? Or how do we see that the tropical moisture contribution is less in Fig 7 than in Fig 6? Please consider providing clearer analyses and plots (e.g., of relative tropical contribution instead).**

Fig 6 and 7 are intended to show the complexity of the 3D structure of the moisture fields from an unconventional perspective, using a snapshot from a 3D viewer. Clearer analyses and plots of the tropical moisture contribution in each case follow suit in Fig8 and 9 (that we will merge into one per some other reviewer's request). These two figures show 2D plots of the percentage of tracer moisture in total precipitable water in the column. Figure captions will be reworded to clarify this point and a mention in the text to better guide the reader, will also be included.

With regard to the "local convergence" mechanism, we were implicitly referring to the study of Dacre et al (2014) about the formation mechanisms of ARs. They assess whether external advection of moisture or convergence of moisture from local sources (evapotranspiration) are the responsible mechanisms for the high amount of total precipitable water defining ARs. We borrow their terminology here, meaning that if high moisture content values are not from external advection, they have to be generated from local moisture, by means of converging mechanisms linked to the dynamics of the front. We will now clarify this in the text as follows:

"The high moisture values behind the front are not related to tropical advection, thus generated by convergence of moisture from local sources occurring along the frontal region"

**• P8.L33-P9.L8 and Figure 10: As LLJ is a key issue investigated, would it not be more informative to calculate and plot how often and when the maximum moisture actually coincides with the low level jet? Also, is LLJ simply defined as the "maximum in wind speed at lower levels" – or is there a wind speed threshold as well? If this is the definition, Fig 10b also shows maximum wind speeds spots at 1 km height, which coincide more with maximum moisture content – why are these spots not identified as LLJ?**

The LLJ is defined as the local maximum in wind speed at low levels (around 1km elevation from the surface) along the cold frontal boundary. There is no threshold criteria. Sometimes there is more than 1 local maximum at low levels, like in Fig 10b, and we made a mistake by placing the label where it is. The LLL is the maximum at around 1km right next to the cold frontal boundary, which is to the west.

It was not our intent to make the LLJ the focus of the paper. As explained earlier, one of our points is that we do not think that ARs can be identified with LLJs, as it is implicit in the current draft definition for the Glossary of Meteorology. We believe that this is sufficiently evident in the cross sections that we show, where the high tropical moisture content, responsible for most of the precipitation, lies in low levels (below 3km) along a relatively narrow zone in the prefrontal region of the system. Just not only within the LLJ. We will tone done down the discussions on the LLJ to give more relevance to the investigation on the origin of moisture in the studied ARs, which is really our main objective in this paper.

**Technical corrections**

• **Figure 5. What is the area highlighted in blue?**

The area highlighted in blue is the domain of simulation. The area highlighted in red is the area of the domain of simulation where moisture is tagged. We will clarify this throughout the text.

• **Figure 11. Please label the vertical axes. Also, should the two axes scale really be identical? If not, consider changing the Ratio axis colour to red.**

We will add labels to the axis in the next version of the paper. The two axes scale is identical, even though the units for each variable are not. This is now going to better clarified in the caption.

• **P7.Eq1: Please define sfc.**

Sfc will be define as "surface"

---

## Author Response (AR1)

Dear Editor Dr. Miralles,

First of all, I would like to thank you for the positive evaluation of our article, as well as for the valuable editorial process which helped to improve the manuscript substantially.

Please, find attached the latest version of the manuscript. We have followed all your advices, as well as all of the proposed by the three reviewers. All the changes are highlighted in red. Together with minor and technical changes, we have made the modifications listed below:

- We have made substantial changes in the Introduction, in order to improve the readability of the text, as well to clarify the transmitted ideas. Particularly, we have increased the clarity of the discussion regarding the state of the art in the differentiation between the low-level jet and the warm conveyor belt.

- We have divided de "Data and Methods" sections in three subsections; Data, The Eulerian Tracer Tool and Methods. We have substantially increased the description of the tracers tool.

- We have unified Figures 8 and 9. Additionally, we have removed Figure 2, which added recurrent information.

Kind Regards,

Jorge Eiras-Barca, MSc.

---

## Author Response (AR2)

We would like to thank the editor for his help and positive evaluation of our work. Please, find below the list of all his concerns addressed one by one.

**– Abstract: 'Vertical cross sections of the moisture (content?) suggest'.**

*"content"* is going to be added to the final version of the manuscript.

**– Low-level jet (LLJ) for 'Low-Level Jet (LLJ)'**

"Low-level jet (LLJ) is going to be replaced by "*Low-Level Jet (LLJ)"*.

**– Remove the added 'have' at 'Guan and Waliser (2015) have developed a global detection method'. Specially if this follows as 'More recently, Eiras-Barca et al. (2016) proposed' (in past tense).**

The "have" located in P2L18 is going to be removed in the final version of the manuscript.

**– 'Into the mid latitudes' for 'into mid latitudes'.**

"Into the mid latitudes" located at P2L21 is going to be replaced by *"into mid latitudes".*

**– There are over 20 acronyms in the article. Please revise if you could be a bit more benevolent to the readers; especially to those that do not have a meteorology background. Perhaps acronyms such as TME (used three times), SLP (used only once and undefined), or TCS (only used in two captions) could be spelled out.**

Following the advice of the editor, we are going to keep the acronyms listed below:

- IWV
- IVT
- WCB
- WRF
- LLJ
- AR

The rest of the acronyms in the current manuscript are going to be spelled out in the final version of the manuscript.

**– Correct 'Avelino, A. and Dall?erba, S.'.**

"Avelino , A. and Dall?erba, S" is going to be replaced by "*Avelino , A. and Dall'erba, S*" *in the final version of the manuscript.*

**– Revise grammar at '…with the uncertainty in the "real word" is due to the WRF model error'.**

In this paragraph there are a couple of words that were wrongly typed in, so that the sentences didn't make sense. One is that caught by the editor, where "with" should read "while", and there is another at the beginning, in the sentence "The strategy consists in replicating the prognostic equations for the

different moisture species IN equations for moisture tracers.", which should be "The strategy consists in replicating the prognostic equations for the different moisture species WITH equations for moisture tracers.", Both typos are going to be corrected.

**– 'WRF Single-Moment 6-Class Mycrophysics Scheme (WSMC6) microphysics scheme'. The second 'microphysics scheme' is not needed. The first use of 'microphysics' has a typo.**

'WRF Single-Moment 6-Class Mycrophysics Scheme (WSMC6) microphysics scheme' will be replaced by '*WRF Single-Moment 6-Class mycrophysics scheme (WSMC6)*'.

**– Section 2.3 should have a more specific title than 'Methods', since the tracer tool is also a method. Maybe 'WRF simulation settings', or similar.**

Section 2.3 is going to read as '*WRF simulations setup*' in the final version of the manuscript.

**– 'good quality statuin' (?)**

"good quality statuin" is a typo which is going to be replaced by "*good quality stations*" in the final version of the manuscript.

**– 'despite the fact that precipitation is known to be the most difficult parameter'. Precipitation is not a parameter. Also add 'arguably' somewhere; now the statement feels awkwardly categorical.**

The full sentence is going to read '*However, despite the fact that precipitation is arguably the most difficult variable to simulate in a numerical model (...)*"

**– 'The reason for the latter is shown in Figure 1' for 'The reason is shown in Figure 1'.**

"latter" is going to be removed from the final version of the manuscript.

**– '90% of the precipitable water in some points', change 'points' for 'areas' or similar.**

The complete sentence is going to be reworded as follows:

"(...) accounts for about 80%-90% of the precipitable water and locally exceeding this contribution."

**– Correct 'montainous'.**

"montainous" is going to be replaced by *"mountainous"*.

**– 'Figure 7.b shows 24h-accumulated percentage of precipitation'. Add 'the' before '24'.**

"*the*" is going to be added before "24".

**– Missing space at 'per day(Buishand...'.**

The cited space is going to be added.

**– The colors in Figure 5 and 6 are still not reflected in the colormap. Please fix.**

We are now going to add a colorbar for subfigures a and b (which was the same colorbar used in subfigures c and d but without attenuantion). Please, note that level height in terrain can be referenced by vertical axis in km.

[Figure]

– Correct 'water stored vapor'.

"water stored vapor" is going to be replaced by *"stored water vapor"* in the final version of the manuscript.

– 'a larger amount of cases' for 'more cases'.

'a larger amount of cases' is going to be replaced by *"more cases"* in the final version of the manuscript.

– Correct the sentences that are repeated in the last paragraph of the Conclusion.

Repeated sentences in the last paragraph of the conclusions are going to be rewritten, and the entire paragraph is going to read as follows:

*"Finally, our findings suggest that the maximum of tropical moisture does not necessarily coincide with the LLJ of either extratropical cyclone analyzed . Instead, this maximum is located in near surface*

*levels at lower latitudes to gradually ascend in northern latitudes, but still remaining below 2 km, mostly within the boundary layer, in contrast with findings in other studies (Dacre et al., 2014). The maximum of tropical moisture may be situated below and toward the back, or ahead the LLJ, which is located along the cold front. Both events are clear examples of ARs from the point of view of vertically integrated variables, such as IWV and IVT used in most detection algorithms; however the vertical distribution of moisture of tropical origin reflects the complex processes leading to precipitation. The new 3D tracer tool will allow us to delve into these processes and explore the role of Tropical Moisture Exports in the initiation and intensification of AR events."*

**Finally, I still feel that the text in Results and Discussion is rather succinct (less than 1000 words). This means the core of the paper will span less than one page (excluding figures) in the final manuscript (one page is around 1200 words). While this is not an important issue, it does not seem to scale with the number of figures and the amount of information they contain.**

Our intent with this paper is to make a concise contribution on the origin of moisture in two AR events, underscoring the importance of the tropical source, which is often not fully acknowledged. This is perhaps because of the lack of tools for moisture tracking as precise as the eulerian water vapor tracers.. We think that the article highlights both points about the power of the tracer tool and the importance of tropical moisture in ARs nicely, as it is, We agree with the editor in the fact that there is still much to be done interpreting and discussing results, but in this paper, since it is about two cases only, we prefer to keep the message short, and leave general conclusions for a study with a larger number of events and higher resolutions. We are already starting to address this issue, so in further articles we are going to be able to provide more technical and specific discussions.

Kind Regards,
Jorge Eiras-Barca.